



# Assessing technical aspects of ocean alkalinity enhancement approaches

Matthew D. Eisaman[1]*, Sonja Geilert[2]*, Phil Renforth[3]*, Laura Bastianini[3], James Campbell[3], Andrew W. Dale[4], Spyros Foteinis[3], Patricia Grasse[4,5], Olivia Hawrot[3], Carolin R. Löscher[6], Greg H. Rau[7,8], Jakob Rønning[6]

[1]. Department of Earth & Planetary Sciences and Yale Center for Natural Carbon Capture, Yale University, New Haven, CT, USA
[2]. Department of Earth Sciences, Utrecht University, Utrecht, The Netherlands
[3]. Research Centre for Carbon Solutions, Heriot-Watt University, Edinburgh, EH14 4AS, United Kingdom
[4]. GEOMAR Helmholtz Centre for Ocean Research Kiel, 24148, Kiel, Germany
[5]. German Centre for Integrative Biodiversity Research (iDiv) Halle-Jena-Leipzig, 04103, Leipzig, Germany
[6]. Danish Institute for Advanced Study (DIAS)/Nordcee, University of Southern Denmark, Odense, Denmark
[7]. Planetary Technologies, Inc., Dartmouth, NS, Canada
[8]. Institute of Marine Sciences, University of California, Santa Cruz, CA, USA

*Correspondence to*: Matthew D. Eisaman (matthew.eisaman@yale.edu), Sonja Geilert (s.geilert@uu.nl), Phil Renforth (p.renforth@hw.ac.uk)

**Abstract**

Ocean alkalinity enhancement (OAE) is an emerging strategy that aims to mitigate climate change by increasing the alkalinity of seawater. This approach involves increasing the alkalinity of the ocean to enhance its capacity to absorb and store carbon dioxide ($CO_2$) from the atmosphere. This chapter presents an overview of the technical aspects associated with the full range of OAE methods being pursued and discusses implications for undertaking research on these approaches. Various methods have been developed to implement OAE, including: the direct injection of alkaline liquid into the surface ocean, dispersal of alkaline particles from ships, platforms or pipes, the addition of minerals to coastal environments, or the electrochemical removal of acid from seawater. Each method has its advantages and challenges, such as scalability, cost-effectiveness, and potential environmental impacts. The choice of technique may depend on factors such as regional oceanographic conditions, alkalinity source availability, and engineering feasibility. This chapter considers electrochemical methods, the accelerated weathering of limestone, ocean liming, the creation of hydrated carbonates, and the addition of minerals to coastal environments. In each case, the technical aspects of the technologies are considered and implications for best-practice research are drawn. The environmental and social impacts of OAE will likely depend on the specific technology and the local context in which it is deployed. Therefore, it is essential that the technical feasibility of OAE is undertaken in parallel with, and informed by, wider impact assessments. While OAE shows promise as a potential climate change mitigation strategy, it is essential to acknowledge its limitations and uncertainties. Further research and development are needed to understand the long-term effects, optimize techniques, and address potential unintended consequences. OAE should be viewed as complementary to extensive emission reductions, and its feasibility may be improved if it is operated using energy and supply chains with minimal $CO_2$ emissions.



## 3.1 Introduction

### 3.1.1 Overview of Ocean Alkalinity Enhancement approaches

The oceans could be artificially alkalized by the addition of alkali (sodium, Na, or potassium, K) and alkaline (magnesium, Mg, or calcium, Ca) silicates, carbonates, oxides, and hydroxides, often as solids, but also in dissolved, aqueous form. The
suite of technologies that aim to achieve this is referred to as ocean alkalinity enhancement (OAE). Figure 3.1 provides a comparative overview of some of the most widely considered OAE approaches.

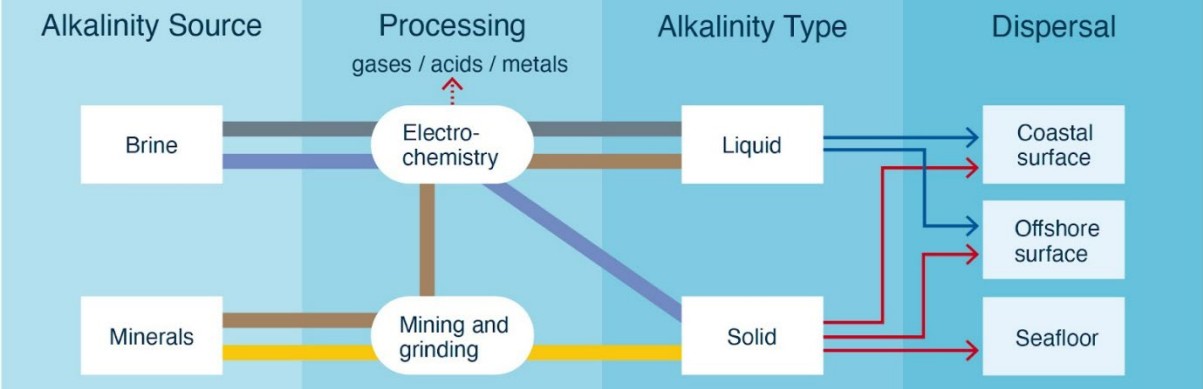

**Figure 1: Categorization of different OAE approaches by alkalinity source, processing method, alkalinity type, and dispersal location. Each pathway colour represents a unique approach. Dispersal location options are determined by alkalinity type. Liquid**
**alkalinity will likely be dispersed on the coast due to its relatively low value of alkalinity per unit volume and mass, whereas solid alkalinity could be dispersed nearshore, offshore, or directly on the seafloor (in shallow water <100 m depth).**

Approximately 0.25 Gt yr$^{-1}$ of carbon is removed from the atmosphere due to weathering of silicate rocks on land (Hartmann et al., 2009). The products of this natural process (e.g., Na$^+$, K$^+$, Mg$^{2+}$, Ca$^{2+}$, HCO$_3^-$, CO$_3^{2-}$, H$_4$SiO$_4$, etc.) are transferred, via rivers and groundwaters, to the ocean where they can durably reside for thousands of years (Renforth and Campbell, 2021).
This natural process can be enhanced by first mining and grinding the Mg- and Ca-rich silicate rocks such as basalt and peridotite into fine powders and adding them to the ocean (Rigopoulos et al., 2018; Köhler et al., 2013). However, for full dissolution of silicate minerals in the ocean surface, very small particles are needed (<10 μm), and as such, the grinding energy to achieve this can be prohibitive (Hangx and Spiers, 2009, Strefler et al., 2018). Instead, larger grains (<100 μm) can be applied to coastal zones (Montserrat et al., 2017) where waves and tides have been suggested to help break down the particles
and accelerate dissolution in seawater in close contact with the atmosphere in a process known as coastal enhanced weathering (CEW) (Chapter 3.6 and 3.7). Although these larger particles dissolve more slowly, they have a lower initial environmental footprint and could be integrated into coastal management schemes such as beach nourishment (Foteinis et al., 2023).

Carbonate rocks, such as limestone (CaCO$_3$) and dolomite (CaMg(CO$_3$)$_2$), are sometimes proposed as an alternative to silicate
rocks for OAE due to their faster dissolution in water. However, the surface ocean waters are almost everywhere supersaturated with respect to calcite and aragonite (Orr et al., 2005), implying that limestone is unlikely to dissolve. One solution is to allow



the $CaCO_3$ to sink to deeper water where it is undersaturated (Harvey et al., 2008), but the significant delay in making contact with the atmosphere and technical challenges limit this approach. Another solution is to convert the limestone to a more soluble form (i.e., calcium bicarbonate: $Ca(HCO_3)_{2(aq)}$) by first dissolving it in a reactor with high p $CO_2$ (Rau & Caldeira, 1999). This

approach is termed accelerated weathering of limestone (AWL) (Chapter 3.3). Potential improvements to AWL include systems such as buffered AWL, whereby hydrated lime $(Ca(OH)_2)$ is added to buffer the unreacted  $CO_2$ before being discharged to seawater (Caserini et al., 2021). Calcium bicarbonate solutions for OAE may also be produced electrochemically (Rau, 2008). For these approaches to be meaningful for CDR, the concentrated  $CO_2$ used in the process must come from the atmosphere, via direct air capture (DAC) or from biomass combustion or metabolism.


Alternatively, limestone could be used to create more reactive materials such as lime (CaO) or $Ca(OH)_2$, which dissolve rapidly in the ocean surface – a process referred to as ocean liming (OL) (Khesghi, 1995) (Chapter 3.4). Other fast-dissolving solids have been suggested as liming agents including brucite $(Mg(OH)_2)$ (Renforth and Kruger, 2013) and sodium carbonate ('soda ash', $Na_2CO_3$) (Khesghi, 1995). However, CaO could be mass-produced by the mining, grinding and then calcining of

limestone, potentially using the pre-existing spare capacity of the cement industry (Renforth et al., 2013). The $CO_2$ produced in the calcination step can be stored geologically or even utilized and the CaO, or most likely the $Ca(OH)_2$, transported and spread to the oceans. Nevertheless, open questions remain, particularly around the effect of localized pH increase on the marine ecosystems in the wake of the delivery vessels (Caserini et al., 2021; He and Tyka, 2023), or pipes, while the potential runaway CaCO3 precipitation could lower the $CO_2$ sequestration efficiency of the approach (Moras et al., 2022).


Alternative pathways are being explored to cost-effectively hydrate minerals and use them as reactive alkaline feedstocks (Chapter 3.5). Ikaite is one example of a hydrated calcium carbonate mineral which is not supersaturated in the ocean, making it potentially viable for OAE (Renforth et al., 2022). In general, hydration of carbonates has the potential to be less energy intensive than calcination of limestone, while offering comparable alkalinity enhancement to lime and slaked lime.


Aqueous salt solutions such as seawater and brines (e.g., desalination wastes, and geological fluids) could potentially provide an abundant source of alkalinity through their electrochemical processing to produce aqueous  $NaOH_{(aq)}$ or other hydroxides, which can be used for near-instant OAE and  $CO_2$ drawdown (Chapter 3.2). There are two main methods of electrochemically generating alkalinity from aqueous salt solutions: electrolysis and electrodialysis. Electrolysis (Willauer et al., 2014) produces

high-concentration (approx. 26 wt.%) $NaOH_{(aq)}$, along with significant quantities of $H2_{(g)}$ and $Cl2_{(g)}$ which must be used within existing energy or product markets or safely stored through reaction with silicate rocks. Electrodialysis, (Eisaman et al., 2012) produces  lower concentration $NaOH_{(aq)}$ (approx. 4 wt.%) along with $HCl_{(aq)}$ and negligible amounts of $H2_{(g)}$ and $O2_{(g)}$ that are vented. Electrodialysis has a lower theoretical voltage drop than electrolysis per mole of alkalinity generated because it relies on enhancing water dissociation and the subsequent separation of H+ and OH- ions across ion exchange membranes, while

electrolysis employs the splitting of water at an electrode surface (Kumar, Du, and Lienhard, 2021). That said, electrodialysis



produces lower concentrations of $NaOH_{(aq)}$ than electrolysis. In electrolysis $H_{2(g)}$ can be burned for energy, or utilized, but the $Cl_{2(g)}$ can be difficult to dispose of and is a potential environmental hazard. In electrodialysis, a use or neutralization pathway must be found for the dilute $HCl_{(aq)}$, for example, by neutralization upon contact with abundant sources of mineral alkalinity.

Even though OAE approaches have the potential to remove atmospheric $CO_2$ at the Gt yr$^{-1}$ scale, they are also responsible for carbon and other emissions during their life cycle. For example, nitrogen-containing explosives (Tovex) that are typically used for mining which can impact eutrophication (Foteinis et al., 2022), whereas nickel (Ni) releases from olivine dissolution could contribute to aquatic toxicity (Foteinis et al., 2023), although low solubility or co-precipitation with secondary minerals may limit the impact (Guo et al., 2022). Therefore, for sustainable and scalable OAE the life cycle environmental impacts of each

approach should be considered and accounted for via life cycle assessment (LCA), rather than simply relying on carbon balances alone. By doing so, not only is the net carbon dioxide equivalent ($CO_{2eq}$) removal quantified but avoided emissions and tradeoffs with other environmental impacts are also identified. For consistent and meaningful LCAs for OAE, standardized guidelines are required, since the relevant ISO standards (ISO 14040 and 14044) only provide generic guidance that is not technology/sector specific. For this reason, the LCA should be specially tailored to OAE applications. Previous work on LCA

best practices for similar sectors, such as those for DACS (Cooney et al., 2022) and for the wider CDR sector (Terlouw et al., 2021) can serve as a useful starting point.

### 3.1.2 OAE Technology Readiness Level

Technology readiness levels (TRLs), developed by NASA in the 1970s, are an estimate of technological maturity. TRLs are based on a scale from 1 to 9, with 9 being the most mature technology (Heder, 2017). Research institutes tend to focus on

TRLs 1 to 4, while the private sector focuses on TRLs 7 to 9. Several OAE approaches lie between TRLs 4 to 7, sometimes called the 'Valley of Death', where neither research institutes nor the private sector prioritize investment.



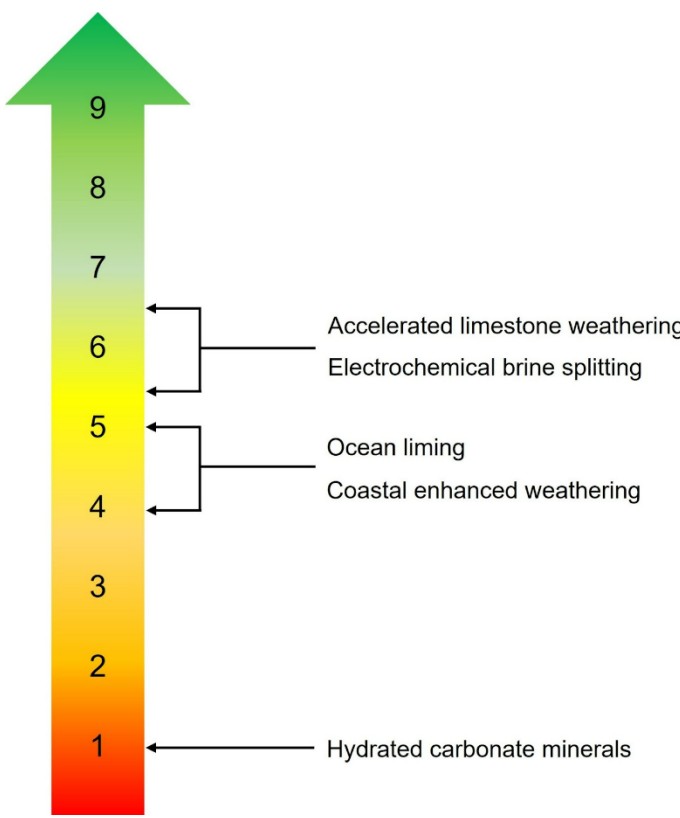

**Figure 2. Technology Readiness Levels (TRLs) for different Ocean Alkalinity Enhancement approaches: 1. Basic principles observed; 2. Technology concept formulated; 3. Experimental proof of concept; 4. Technology validated in laboratory environment;**
**5. Technology validated in a relevant environment; 6. Technology demonstrated in a relevant environment; 7. System prototype demonstration in an operational environment; 8. Actual system completed and qualified; 9. Actual system proven in an operational environment.**

The TRL of OAE approaches is summarized in Fig. 3.2. A feasibility case study for an AWL system attached to a coastal power plant in Taiwan (Chou et al. 2015), and most recently a pilot scale AWL reactor for flue gas separation (Kirchner et al.,
2020), suggests TRL 5 - 6. Electrochemical brine splitting has a similar TRL of around 6 with several start-up companies in the process of deploying pilot demonstrations. CEW has a lower TRL of between 4 to 5, currently undergoing field trials in Southampton, New York to prove its efficacy. There are still significant challenges to scaling up the approach, particularly surrounding monitoring reporting and verification (e.g., Burt et al., 2023, see Chapter 6), potential ecosystem effects (Bach et al., 2019), as well as logistical challenges around mining, grinding, and transporting enough alkaline material from land to
distribute in the marine environment which would require massive infrastructure and long supply chains (Renforth et al., 2013). Previously, OL has been assigned a TRL of 3 to 4 (McLaren, 2012), but can now be considered to have advanced to a TRL of 4 to 5 after recent field trials in Florida. Finally, the production and application of hydrated carbonate minerals such as ikaite has a TRL of 1, currently under investigation at the bench-scale at Heriot-Watt university examining aspects such as air stability and seawater dissolution kinetics.





Overall, while OAE may have great potential for CDR, there are still many unanswered questions about the long-term ecological impacts and the feasibility of implementing these techniques on a large scale. As such, further research and development is needed to increase the TRLs of these approaches and which, if any, other approaches should be scaled up. The ocean is a heterogeneous system and field-tests are required across a variety of oceanic conditions, e.g., temperatures, upwelling velocities, seawater chemistries, and biological profiles. A research program designed to accelerate technology

development and demonstration of pilot-scale facilities will also need to assess any potential ecological impacts and governance issues.

## 3.2 Electrochemical production of alkalinity for OAE

### 3.2.1 Technical Summary - chloride brines

Aqueous brine (for example, $NaCl_{(aq)}$) represents an abundant source from which aqueous alkalinity (for example $NaOH_{(aq)}$)

can be generated using electrochemistry. In these approaches, the alkalinity is in the form of hydroxide ions ultimately derived from the water in the brine stream, with the dissolved brine ions (for example $Na^+$ and $Cl^-$) providing the conductivity and charge balance needed for the process.

The two primary electrochemical processes used to generate alkalinity from brine are electrolysis (O'Brien, Bommaraju, and

Hine, 2005, pp.31-34) and electrodialysis (Strathmann, 2011, pp.163-167), as shown in Fig. 3.3. Electrolysis generates higher concentration alkalinity than electrodialysis but requires a higher electrical potential per mol of alkalinity to do so.

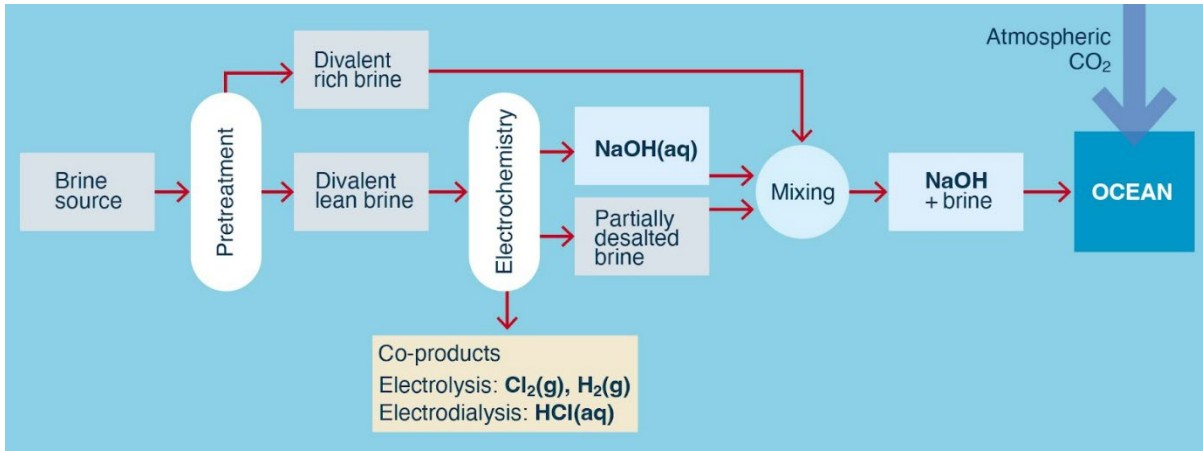

**Figure 3. Process flow for the electrochemical conversion of aqueous chloride brine into alkalinity.**

This is because electrolysis uses more energy-intensive *water splitting* at electrodes to generate the alkalinity, while

electrodialysis uses *enhanced water dissociation* at the junction of the bipolar membranes, combined with ion-selective





separation. While the primary by-products from electrolytic alkalinity generation are $Cl_2$ and $H_2$ gasses, the primary by-product from electrodialytic alkalinity generation is aqueous acid, for example $HCl_{(aq)}$.

The electrolytic generation of alkalinity from an NaCl solution (see Fig. 3.4) is essentially the well-known chlor-alkali process
(O'Brien, Bommaraju, and Hine, 2005, pp. 31-34), where aqueous brine (approx. 26 wt%) and NaOH (approx. 28 wt%) are converted into less concentrated brine (approx. 24 wt%), more concentrated NaOH (approx. 30 wt%), hydrogen gas ($H_2$), and chlorine gas ($Cl_2$) (Kumar, Du, and Lienhard, 2021).

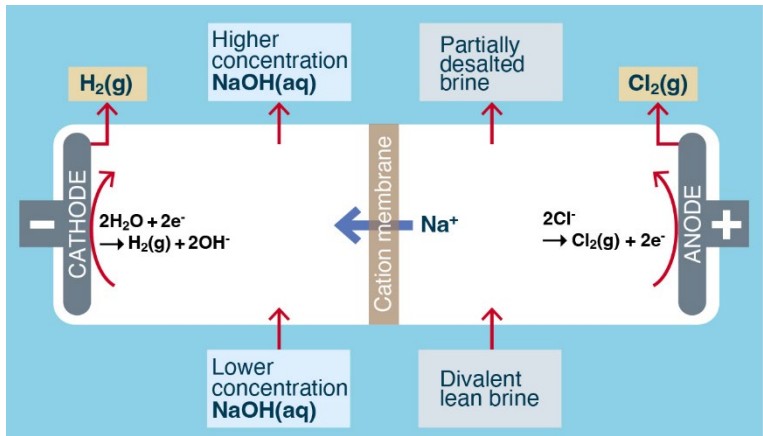

**Figure 4. Typical process flow for the electrolytic conversion of aqueous NaCl-rich brine into alkalinity using the chlor-alkali membrane process. A chlor-alkali diaphragm process also exists but is not shown (Kumar, Du, and Lienhard, 2021).**

The electrodialytic generation of alkalinity from an NaCl brine typically uses three-chamber bipolar membrane electrodialysis (BPMED) (Strathmann 2011, pp.163-167, Fig. 3.5). In this process, aqueous brine (approx. 3.5-5 wt%), HCl (approx. 2 wt%), and NaOH (approx. 2 wt%) are converted into less concentrated brine (approx. 2-3.5 wt%), more concentrated HCl (approx.
3-4 wt%), and more concentrated NaOH (approx. 3-4 wt%). Hydrogen gas ($H_2$) and Oxygen gas ($O_2$) are created at the end electrodes, but in contrast to electrolysis, because there are typically 50 - 200 membrane triplets between each pair of electrodes, the rate of $H_2$ and $O_2$ gas generation relative to the rate of NaOH production is negligible, reduced by a factor of (number of cell triplets)$^{-1}$ relative to electrolysis. In practice the $H_2$ and $O_2$ gasses generated during electrodialysis are combined and vented. The $HCl_{(aq)}$ generated in this process is used on land in processes that result in the neutralization of the acid, for
example in the neutralization of alkaline waste ponds found at sand and gravel operations. Scaling to Gt $CO_2$ y$^{-1}$ will require larger-scale uses of the acid such as the pre-treatment of silicate rocks to enhance the kinetics and capacity of $CO_2$ mineralization (Guy and Schott, 1989; Pollyea et al. 2017).





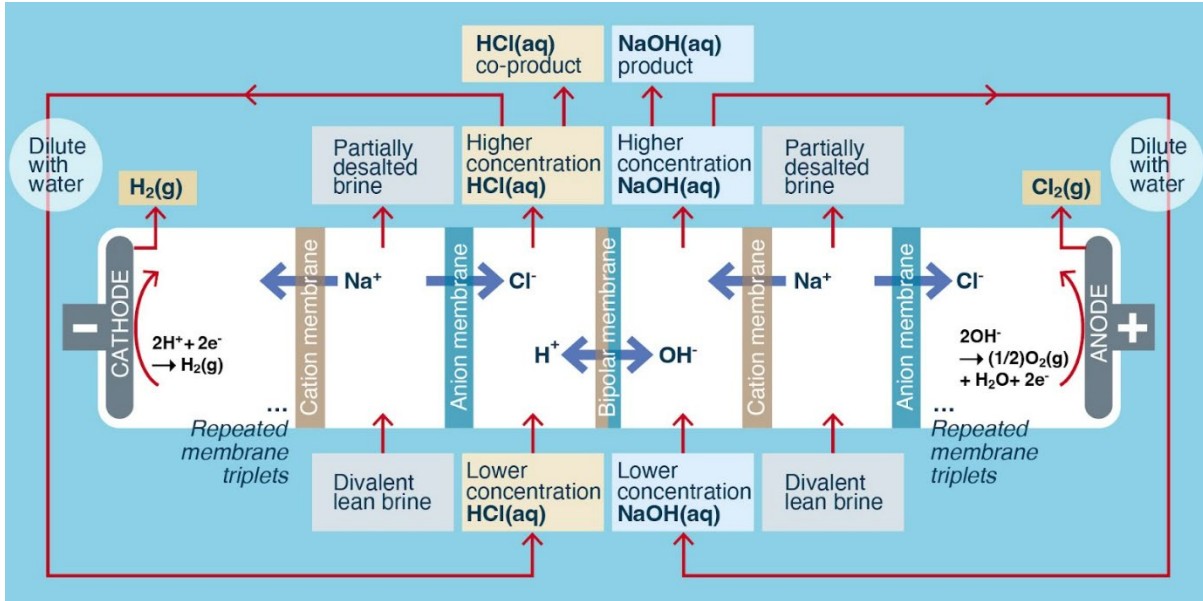

**Figure 5. Typical process flow for the electrodialytic conversion of aqueous brine into alkalinity.**

Once the alkalinity is generated, it must be dispersed into the ocean. An advantage of aqueous hydroxides such as $NaOH_{(aq)}$ is that the rate at which the alkalinity source is *added* to the ocean is equal to the rate at which alkalinity is actually *delivered* to the ocean. In contrast, with solid forms of alkalinity such as crushed minerals, the relationship between rate of alkalinity source *added* and the rate at which this potential alkalinity is *delivered* to the ocean depends on the dissolution kinetics of the solid.

Once the alkalinity has been delivered to the ocean, the response of the ocean and atmosphere is governed by two timescales: an immediate timescale that corresponds to the response of the carbonate chemistry of the ocean (Reaction 1):

$$OH^- + CO_{2(aq)} \rightarrow HCO_3^- \qquad\qquad\qquad \text{(R1a)}$$

$$OH^- + HCO_3^- \rightarrow H_2O + CO_3^{2-} \qquad\qquad \text{(R1b)}$$

and a slower timescale of weeks to months that corresponds the re-equilibration of $CO_2$ in the air and surface ocean via air-sea gas exchange (Reaction 2):

$$CO_{2(air)} \rightarrow CO_{2(aq)} \qquad\qquad\qquad\qquad \text{(R2a)}$$

$$CO_{2(aq)} + H_2O + CO_3^{2-} \rightarrow 2HCO_3^- \qquad\quad \text{(R2b)}$$

The net reaction described by Reactions (1) and (2) once equilibrium has been reached is (Reaction 3):

$$OH^- + aCO_{2(aq)} \rightarrow bHCO_3^- + cCO_3^{2-} + dOH^- + eH_2O \qquad\qquad \text{(R3)}$$





where coefficients $a$ - $e$ are molar ratios relative to the added OH$^-$ (added alkalinity), $a = b + c$ (carbon mass balance) and $b + 2c + d = 1$ (charge balance). For example, modelling in CO$_2$SYS (citation) shows $a = 0.827$, $b = 0.742$, $c = 0.086$, $d = 0.086$, $e = 0.095$ in seawater at an equilibrium pH of 8.1, S = 35 ppt, T = 20 °C and P = 1 bar. These ratios are sensitive to the
preceding seawater variables; previously reported values for the ratio of moles of removed CO$_2$ to moles of added alkalinity ($a$ coefficients) range from 0.75 to more than 0.85 (Tyka et al. 2022; He and Tyka 2023; Renforth and Henderson 2017; Wang et al. 2023).

As shown in Reaction (R1), on fast timescales, the addition of alkalinity decreases the dissolved CO$_2$ concentration, putting
the surface ocean pCO$_2$ out of equilibrium with atmospheric pCO$_2$. On slower timescales of weeks to months for Reaction (R2), equilibrium is re-established as CO$_2$ from the atmosphere replenishes the CO$_2$ deficit in the surface ocean. The combined result of these two processes is the net removal of CO2 from the atmosphere and storage as oceanic bicarbonate and carbonate ions.

Upon dispersal to the ocean, the added alkalinity is increasingly diluted as it moves away from the point of addition. This results in a mixing zone centred at the point of alkalinity addition where the increase in pH and total alkalinity (TA) is largest. The ratio of the rate of alkalinity addition to the rate of dilution must be kept sufficiently low to avoid the precipitation of Mg(OH)$_2$ (which can result in an undesired increase in turbidity) or CaCO$_3$ (which reduces the efficiency of OAE for CO$_2$ removal) within the mixing zone (Hartmann et al. 2023). Due to this constraint and the permitted limits in the mixing zone for
parameters such as pH and turbidity, in practice the pH of aqueous alkalinity may need to be reduced prior to dispersal. For example, prior to release into the ocean, the alkalinity could be mixed with the partially desalted brine stream from which the alkalinity was generated.

To reduce the need for dilution, the alkalinity may first be contacted with CO$_2$ in air to decrease the pH by converting some
of the hydroxide (OH$^-$) into carbonate (CO$_3^{2-}$) (Stolaroff et al. 2008). This has the added advantage that all the CO$_2$ captured in this way is measurable and verifiable through direct measurement. This effectively provides a tuneable knob to perform "partial direct air capture (DAC)" to the degree required to reach the target pH, at which point it can be diluted to the pH required by permitting, and the remainder of the CO$_2$ removal can occur via OAE once it is dispersed into the ocean. The advantage of this use of DAC as a "partial pre-equilibration" for OAE compared to standard DAC, is that when used as a
preparation step for OAE, no energy needs to be applied to release the CO$_2$ as a pure gas. As an example, if one generates approximately 4% NaOH$_{(aq)}$ using electrodialysis, partial DAC can be used to bring the pH into the range 11 - 12, at which point it can be blended with waste brine to the final pH suitable for ocean delivery.

Rather than partial DAC and dilution, aqueous alkalinity may be directly delivered to the ocean at higher concentrations as
long as natural or engineered dilution rates in the mixing zone avoid unwanted precipitation and stay within permitted bounds.





Alternatively, when discharged within permitted turbidity limits, more slowly dissolving forms of particulate alkalinity such as $Mg(OH)_{2s}$ could be used to distribute the added alkalinity more evenly in space and time (Fakhraee, M. et al., 2023).

A process that is very related to OAE using chloride brines is "indirect ocean capture" or IOC (de Lannoy et al. 2018; Eisaman
et al. 2018; Eisaman et al. 2020), also referred to as "direct ocean capture" (DOC) or " $CO_2$ removal from ocean water" (Kim et al., 2023). This approach employs *alkalinity cycling* to remove $CO_2$ from the ocean, but *without a net increase in ocean alkalinity* or DIC. Because the net alkalinity is not enhanced in this process, it should not be labelled as OAE. In one version of this approach, electrodialysis is first used to generate $HCl_{(aq)}$ and $NaOH_{(aq)}$ from brine streams containing $NaCl_{(aq)}$. The acid is used to acidify seawater, decreasing its pH and alkalinity, and shifting all its DIC to dissolved $CO_2$ gas, which is then
vacuum stripped out of the seawater (de Lannoy et al. 2018; Eisaman et al. 2018). The alkaline base is then added to the now decarbonized seawater to restore its lost alkalinity, resulting in $CO_2$ moving from the air to the seawater to restore equilibrium, thereby replacing the vacuum-stripped $CO_2$. At a high level, this approach uses the ocean as a pump, in contrast to OAE, which uses the ocean as a sponge. In a second version of this approach, the $NaOH_{(aq)}$ is added to seawater to remove $CO_2$ as $CaCO_{3(s)}$, with additional $NaOH_{(aq)}$ then added to restore this lost alkalinity and draw $CO_2$ from the air to replace the removed
$CO_2$ (de Lannoy et al. 2018; Eisaman et al. 2018; La Plante et al. 2021). The precipitation of $CaCO_{3(s)}$ releases $CO_2$, making this second version relatively inefficient from a $CO_2$-removal perspective, but may be pursued if other considerations such as ease of verification outweigh this inefficiency.

### 3.2.2 Technical Summary - non-chloride brines and minerals

In addition to the production of alkalinity from chloride salts discussed above, hydroxides can also be electrochemically
produced from non-chloride salt solutions such as Na, K, Ca or Mg sulphates, nitrates or carbonates. One disadvantage of such an approach is that such salts are much less naturally abundant or less soluble than chloride salts, though they can be present in waste streams. However, as will be described, they can be produced from mineral sources of metal cation and recycled anions. Because of electrochemical issues with nitrate salts and because carbonate salts present more limited net carbonation potential and often have less solubility, the focus here will be on metal sulphate salts.



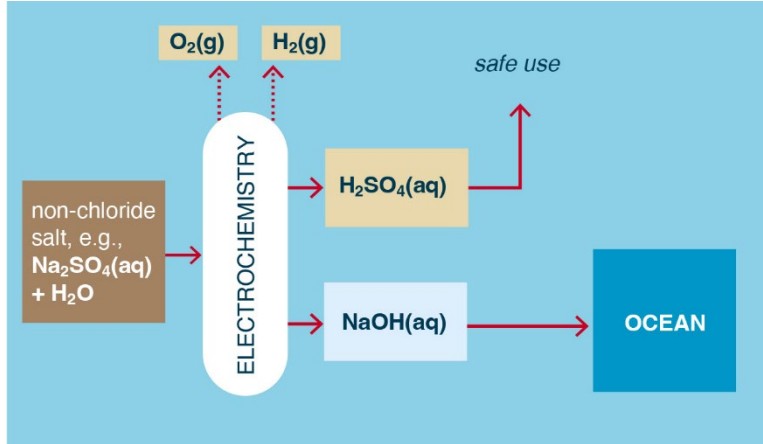


**Figure 6. Example of a non-chloride metal salt used to electrochemically produce an acid, a metal hydroxide, and in the case of electrolysis, H2 and O2.**

As in the case of NaCl, $Na_2SO_4$ solutions can be electrolyzed or electrodialyzed to produce, in this case, $H_2SO_4$ (sulfuric acid) generated/concentrated at the anode and NaOH at the cathode (Little et al. 2012). In electrolysis, $H_2$ and $O_2$ (rather than $Cl_2$)

are also produced at the anode and cathode, respectively (Fig. 3.6).

While the NaOH can be used for OAE CDR as described above, uses of the acid at the production scales required for globally significant OAE must be identified. Such acids (including the hydrochloric acid described in the previous section) can be reacted with alkaline minerals to produce more neutral metal salts and water. For example, the reaction of sulfuric acid with

the silicate mineral forsterite ($Mg_2SiO_4$) yields $MgSO_4$, $SiO_2$ and $H_2O$ (Reaction 4):

$$Mg_2SiO_4 + 2H_2SO_4 + SiO_2 + 2H_2O \qquad \text{(R4)}$$

As suggested by House et al. (2007) such metal salts produced from the preceding reaction are in theory benign and could be

added to the ocean. However, most silicate minerals contain multiple metals that upon acidification yield metal salts such as Mg, Ca, Fe, Ni, Co and Na sulphates or chlorides in solution. While at least some of these metal salts will have limited solubility in alkaline seawater, their disposal in the ocean would be problematic due to potential biological effects (see Chapter 4). One alternative is to take advantage of the differences in the reduction in metal solubility as pH rises to selectively remove the less soluble metals as solid metal hydroxides, such as $Fe(OH)_2$, $Ni(OH)_2$ and $Co(OH)_2$, as is commonly done in metal extraction

from rocks (Hamilton et al. 2020).  Some of the produced NaOH could be used to facilitate pH elevation of the metal salt solution and the resulting valuable metal precipitates can be harvested for further refining. The remaining, more soluble metal salts, e.g., $MgSO_4$, $CaSO_4$ and $Na_2SO_4$, could then provide a more benign way to dispose of the products of acid neutralization.



However, such schemes (e.g., Rau et al. 2013) require loss of $SO_4^{2-}$ (or $Cl^-$ in the case of HCl use) and thus a continuous supply
of (expensive) sulphate would be required. To overcome this challenge, the sulphate can be recycled first as an acid and then
as a metal sulphate and back again (Lammers et al. 2023). For example, a monovalent salt solution, e.g., $Na_2SO_4$, can be
electrolyzed or electrodialyzed to generate $H_2SO_4$ that is again used to leach metal salts from minerals, but where the NaOH
produced in the catholyte is used exclusively to precipitate less soluble polyvalent metals as metal hydroxides (Fig. 3.7), with
the then reformed $Na_2SO_{4(aq)}$ recycled as brine/electrolyte. In this way $Na_2SO_4$ is (largely) conserved and the metal hydroxide
precipitates could then be used as an alkalinity source for OAE if they are at least partially soluble in seawater.

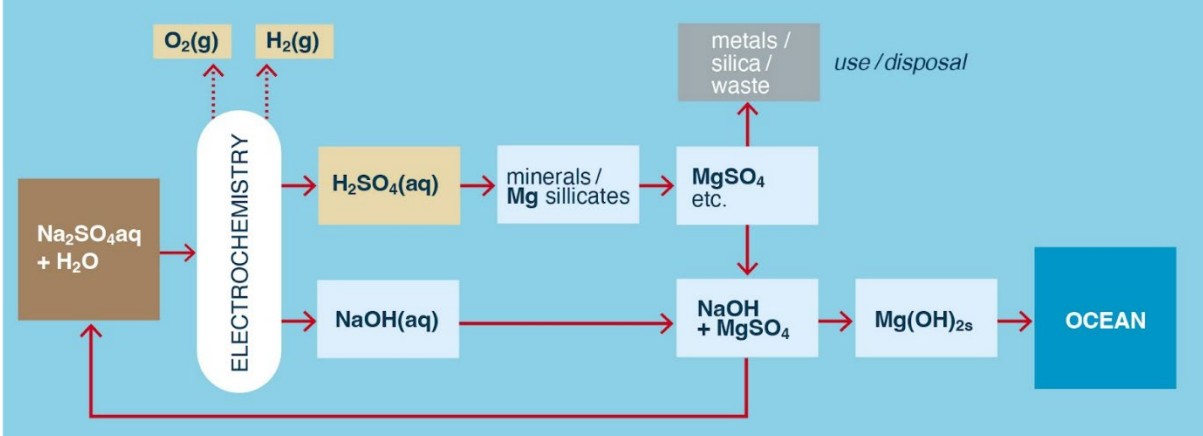

**Figure 7. Example of the indirect production of solid, polyvalent metal hydroxides from minerals and water using electrodialysis or electrolysis of a monovalent salt. H2 and O2 are also produced.**

Another alternative is to use dissolved metal sulphates produced from the mineral/acid leaching directly as electrolyte/brine in
cells where the subsequent precipitation of metal hydroxide inside the cell is avoided or otherwise accommodated (Fig. 3.8).
This could include continually harvesting e.g., $Mg(OH)_2$ or $Ca(OH)_2$ precipitated on or near the cathode (Pan et al. 2018, Sano
et al. 2018) and/or using diaphragms (Kelland et al. 2022) or membrane-less cells (Talibi et al. 2017) to avoid membrane
fouling by precipitates. Compared to the process in Fig. 3.7, this method more directly generates hydroxides from mineral
sources and water.





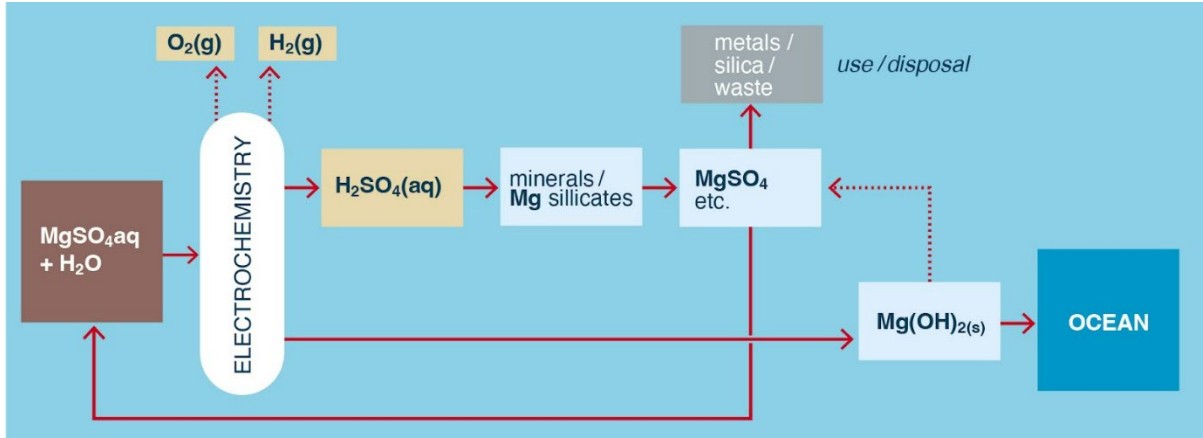

**Figure 8. Example of the direct use of metal salts acid-leached from minerals in the electrochemical production of alkalinity for OAE.**

### 3.2.3 Considerations for best research practices

In this section, we highlight key parts of the brine-to-alkalinity OAE process where the application of best practices is especially critical to performing reproducible research.

*Brine treatment:* The aqueous brine source input into the electrochemical unit for the generation of alkalinity is often seawater, reverse osmosis concentration (ROC), or some other brine stream that also contains the divalent cations $Ca^{2+}$ and $Mg^{2+}$. Because the electromechanical systems used for this purpose have compartments with high pH values (> 13), care must be taken to avoid the precipitation of solid $Mg(OH)_2$ and/or $CaCO_3$ within the electrochemical unit, as this can lead to increasing voltages, current shunting, and increased hydraulic pressure. Some combination of pre-treatment to decrease the concentration of divalent cations and periodic, acidic clean-in-place flushing can avoid this problem. Examples of pre-treatment methods include, but are not limited to, ion exchange and the use of precipitation softening via $NaOH_{(aq)}$ addition.

*Alkalinity generation:* In practice, the generated alkalinity may have a more complex ionic composition than the simplified description provided above, depending on the input brine composition and the properties of the electrochemical system such as membrane permittivity and selectivity. For example, because seawater contains $K^+$ ions and because cation exchange membranes allow $K^+$ and $Na^+$ transport, the alkalinity generated from the electrodialytic processing of seawater will not be pure $NaOH_{(aq)}$ but will also contain some fraction of $KOH_{(aq)}$. This means that in-line measurement proxies of the generated alkalinity such as conductivity and pH should be calibrated by offline measurements such as TA titrations and inductively coupled mass spectrometry (ICP-MS). Sampling ports built into the electrochemical system are recommended for this purpose.

*Aqueous alkalinity dispersal:* As previously mentioned, the ratio of the rate of alkalinity addition to the rate of dilution must be kept low to avoid the precipitation of $Mg(OH)_2$ or $CaCO_3$ within the mixing zone. The carbonate chemistry and turbidity



should be continuously monitored near the point of dispersal. Precipitation will manifest as a decrease in TA and increase in turbidity (see Chapter 4).

*Energy cost,  CO₂ emissions, and economics:* In order for research on electrochemical  OAE to be relevant to CDR performed at globally relevant scales, it is necessary to document or estimate the energy use,  CO₂ emissions, capital costs and operating

costs that are incurred in small-scale systems, and particularly how these would scale for larger systems. These are essential for making informed decisions regarding future RD&D allocations and ultimately decisions about when and where the most cost-effective methods might be deployed.

Environmental and societal impacts and benefits: So as to better inform decision makers, researchers must assess how land,

air, ocean and societal systems might be affected by electrochemical OAE. This includes the environmental and societal consideration of: (i) Land and resource use such as mineral/salt/brine/water extraction, transportation, processing and refining; (ii) The footprint of the facility and its operation; and (iii) The downstream impacts/benefits of the products. Researchers should aim to produce a comprehensive budget of all fluxes across the system boundaries (inputs and outputs of energy and matter) to enable this assessment (see Chapter 5).

**3.3 Accelerated Weathering of Limestone as an OAE Strategy**

**3.3.1 Technical summary**

In a process called Accelerated Weathering of Limestone (AWL), calcium carbonate (derived from carbonate-bearing rocks, e.g., limestone) can be spontaneously carbonated in the presence of elevated p CO₂ and seawater to form predominantly calcium bicarbonate ions in seawater (Rau and Caldeira 1999) via the reaction (Reaction 5):


$$CaCO_{3(s)} + aCO_{2(aq)} + bH_2O \rightarrow Ca^{2+} + cHCO_3^- + dCO_3^{2-} + eOH^- \qquad (R5)$$

where the molar quantities relative to CaCO₃ are approximately: a = 0.65, b = 0.74 , c = 1.48, and d = 0.17, e = 0.18 when re-equilibrated with typical seawater at a p CO₂= 420 µatms. The preceding quantities are halved when expressed in units of

moles per mole of alkalinity since 1 mol of CaCO₃ = 2 mols alkalinity. This implies a maximum tonnes  CO₂ removal per tonne CaCO₃ of about 0.29 or a minimum requirement of about 3.5 tonnes of CaCO₃ per tonne  CO₂ captured and stored.

While AWL is an OAE scheme, given the requirement of elevated  CO₂ to spontaneously drive reaction 5, it has been more widely considered as a   CO₂ emissions reduction technology, analogous to CCS, at coastally located, fossil-fuelled power

plants (Rau and Caldeira 1999, Caldeira and Rau  2000, Rau et al. 2007, Langer et al. 2009, Rau 2011, Haas et al. 2014, Chou





et al. 2015, Kitchner et al. 2020a and b, Caserini et al. 2021, Xing et al. 2022). However, this approach can be relevant to CDR if the concentrated $CO_2$ used is from: i) emissions from biomass respiration, energy (electricity) production, gasification or fermentation, ii) direct air capture, iii) natural emissions from hydrothermal or geothermal activity or iv) possibly natural or artificial upwelling of deep seawater whose p $CO_2$ is high enough and a $CaCO_3$ saturation state low enough to facilitate $CaCO_3$
dissolution.

The $CO_2$ must be of sufficient concentration so that when equilibrated with water or seawater, $CaCO_3$ undersaturation in the solution is affected and the reaction can proceed. Calculations using $CO_2SYS$ (Pierrot 2006) suggest that surface seawater equilibrated with a p $CO_2$ greater than about 2500 µatms is required for $CaCO_3$ undersaturation to occur and for the reaction
to spontaneously proceed. Sufficiently elevated p $CO_2$ drives down solution pH and thus $[CO_3^{2-}]$ to achieve a $CaCO_3$ saturation state that is corrosive to $CaCO_{3s}$. The higher the p $CO_2$ the lower the pH, $[CO_3^{2-}]$, and $CaCO_3$ saturation state ($\Omega_{cal}$) and hence the faster the kinetics of the reaction, the greater the areal and volumetric reaction rates achieved, and the higher the DIC concentration attained. Experiments have shown reaction rates ranging from about $10^{-7}$ to $10^{-5}$ mols m$^{-2}$ of mineral surface sec$^{-1}$. Since volumetric reaction rates are sensitive to carbonate mineral surface area per reaction volume, the interplay among
carbonate particle size, seawater and gas contacting, and flow rates dictate reactor design, size and performance (Rau 2011, Kirchner et al. 2020a, Xing et al 2022).

Once the calcium bicarbonate+carbonate ions are formed and discharged into the ocean it is presumed that the longevity and security of the storage will be equivalent to that of the existing alkaline C in the ocean, on the order of 100,000 years
(Middelburg et al. 2020). This assumes that the AWL reaction (R5) will not be reversed prematurely by enhanced biotic or abiotic $CaCO_3$ precipitation. Biotic calcification in some marine taxa has been shown to increase with increasing alkalinity and rising calcium carbonate saturation state, $\Omega_{cal}$ (Albright et al. 2016, Renforth and Henderson 2017, Gore et al. 2019). Note that calcium carbonate saturation state will be more sensitive to the addition of Ca (bi)carbonate than non-Ca alkalinity since both $Ca^{2+}$ and $CO_3^{2-}$ are being added: $\Omega_{cal} = [Ca^{2+}]\,[CO_3^{2-}]/K_{sp}$ where $K_{sp}$ is a temperature, salinity, and pressure sensitive
solubility constant for calcite. Thus, on a per mole basis, the threshold for carbonate precipitation will be more rapidly reached with calcium-based alkalinity addition relative to the addition of other forms of dissolved metal (bi)carbonates (Fig. 3.9).

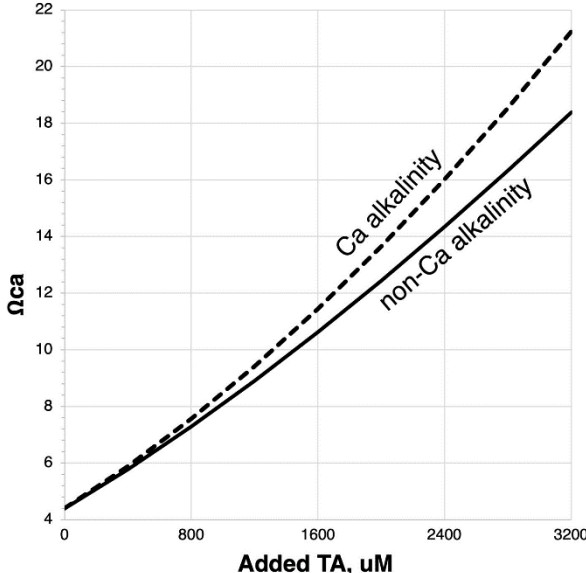

**Figure 9. Ωca(= the saturation state of calcite) response to calcium-based vs non-calcium-based alkalinity added to seawater initially containing 2350 μM total alkalinity (=TA) and equilibrated with a p CO₂ of 420 μatms. Modelled using  CO₂SYS (Pierrot 2006) 380  modified to account for variable  [Ca2+].**

An additional feature that will further promote carbonate precipitation is the degassing of the excess  $CO_2$ from the carbonated solution once exposed to air. This in effect removes acid from the carbonated solution, raising pH, $[CO_3^{2-}]$ and $\Omega_{cal}$. An example of the chemical sequence of events in carbonating seawater using AWL and then re-equilibrating the carbonated seawater with air is shown in Fig. 3.10.



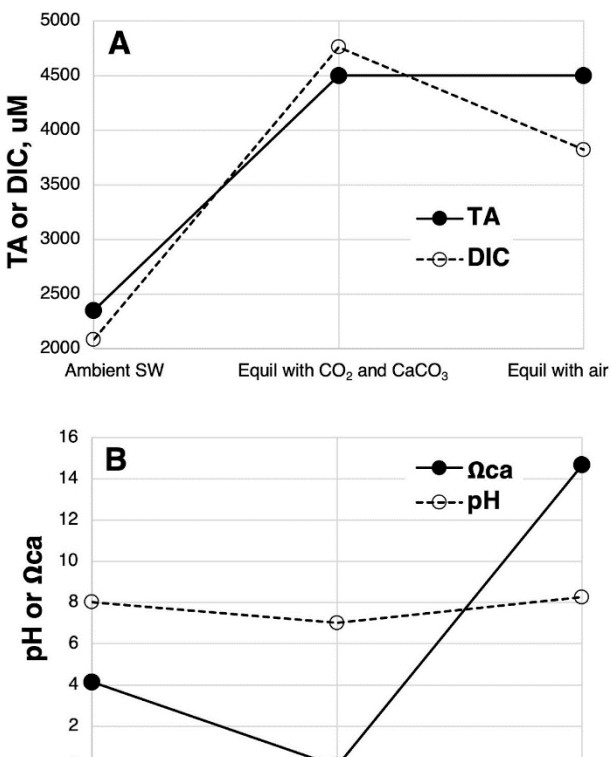

**Figure 10. Example of the chemical progression of AWL starting first with representative, ambient seawater at p $CO_2$=420 µatms, followed by equilibration with $CaCO_3$s and a p $CO_2$ of 10,000 µatms followed by re-equilibration with air p $CO_2$=420 µatms. A) DIC = total dissolved inorganic carbon; TA = total alkalinity. B) $\Omega$ca= the saturation state of calcite. Modelled using $CO_2$SYS (Pierrot 2006) modified to account for variable [$Ca^{2+}$].**


Here the net addition of 2150 µM of dissolved calcium (bi)carbonate in equilibrium with an excess of $CaCO_3$ and a p $CO_2$ of 10,000 µtams yields a transfer of carbon from the gas to seawater of 2679 µM, about 310 µM of which is in the form of $CO_{2(aq)}$. If this solution was then added to the surface ocean and allowed to re-equilibrate with air p $CO_2$=420 µatms, pH would rise above that of the initial seawater (by about 0.24 units) and stored carbon would decline from its high value following carbonation by about 35%, resulting in a net C storage from the AWL process of 1739 µM C (Fig. 3.10A). In a full-scale

facility, Kirchner et al (2020b) modelled a 50% loss of captured carbon following an AWL discharge. The percentage of captured $CO_2$ lost will very much depend on the p $CO_2$ of the gas used as well as the degree of chemical equilibration achieved among gas, carbonate and seawater. Use of high $CO_2$ and its equilibration with seawater, but with incomplete equilibration with carbonate minerals will result in an AWL discharge solution with high DIC largely as $CO_{2(aq)}$, but with relatively little additional alkaline C formed (little long-term C storage).


     Upon equilibration with air, the loss of excess $CO_2$ from the above alkalinized seawater forces a rise in $\Omega_{CA}$ to 14.5 in the above example (Fig. 3.10B). This is >3X higher than the initial ambient seawater and could cause spontaneous $CaCO_3$



precipitation from seawater. However, such an effect is likely countered by the rapid dilution of the carbonated solution with ambient seawater relative to the slow kinetics of $CaCO_3$ precipitation (Fig. 7 in He and Tyka 2023). As long as dilution occurs faster than $CaCO_{3(s)}$ formation, $CaCO_{3(s)}$ precipitation and alkalinity loss can be avoided (He and Tyka, 2023).

The limitations of this approach in the context of OAE CDR include the need for a concentrated non-fossil $CO_2$ source in close proximity to seawater and carbonate minerals. Potential mineral carbonate sources include globally abundant limestone as well as less abundant dolomite and magnesite. Waste marine shell material or carbonate sands can also be considered, especially because aragonitic shell material should be more soluble than calcite (e.g., limestone) and that this aragonite dissolution simply speeds up the natural return of its marine-derived constituents to seawater. Proximity to the ocean is also a requirement both for the water used for carbonation and well as for discharge and storage of the carbonated, alkalized seawater. Also, considering a possible upper limit of only about 25 mg C stably stored/L of seawater, significant pumping of seawater to facilitate gas/carbonate contacting and conversion must occur. Mining, processing and transportation of >3.5 t $CaCO_3$ t$^{-1}$ CDR also needs to be considered as does the size and capital and operating costs of the seawater/ $CO_2$/carbonate contactor. For one AWL design, Xing et al. (2022) estimate an energy cost of 5.7–8.2 GJ and a land requirement of 7.1–13.1 m$^2$ per tonne of $CO_2$ captured and stored, after allowing for degassing of the carbonated seawater following discharge in the ocean.

Advantages of the process include: i) spontaneous, exothermic conversion and long-term storage of $CO_2$, ii) ocean alkalization and thus restoration of ocean pH, iii) relative ease of verifying CDR by carbonating alkalinity prior to release and quantifying the increase in carbon concentration in solution prior to release rather than having to verify CDR occurring in the ocean, and iv) providing a relatively simple, low-tech, widely applicable approach to OAE at coastal sites.

In sum, considering the global abundance of concentrated $CO_2$ waste streams, calcium carbonate mineral resources (including massive waste piles (Langer et al. 2009)) and the reactivity of these minerals in elevated $CO_{2aq}$ solutions, AWL seems an effective way to perform relatively safe, low-cost, low-tech OAE at scale, especially considering that it is routinely used at small scales to alkalinize saltwater aquaria (Huntington 2002) and considering that such spontaneous rock/water/ $CO_2$ reactions provide the primary source of alkalinity naturally present in the global ocean (Middelburg et al. 2020). Its use in CDR is, however, more restricted considering that the $CO_2$ used must be concentrated above that in air in order to make the CDR rate relevant on human timescales. Further research is needed to better determine the desirability, effectiveness and capacity of AWL.

### 3.3.2 Considerations for best research practices

*Purity of feedstocks* Any impurities in the feed $CO_2$ or carbonate mineral has the potential to be released with the discharge of the carbonated seawater. The quantity and impacts of these impurities need to be measured to assess the potential for





downstream environmental impact as well as reduction in reactivity per mass of mineral. For example, Kirchner et al (2020a) found measurable trace metal concentrations in their discharge originating from the limestone used, but all concentrations they considered were "below levels of environmental concern". Only slightly elevated $NO_3^-$ was observed in the discharge seawater that originated from the $NO_x$ from the flue gas processed.


*Monitoring Reporting and Verification.* Because AWL carbonation likely occurs before addition to the ocean, the quantity of carbon captured and converted for long term storage can in theory be easily quantified as an increase in seawater DIC at point of discharge. This is especially true if the carbonated seawater is equilibrated with air prior to release to the ocean so that excess dissolved $CO_2$ (that will ultimately be lost from seawater once released) is not counted as sequestered carbon (Fig. 2).

Otherwise, the net $CO_2$ removed and stored can be measured/ calculated by either: i) bubbling with air a subsample of the freshly carbonated seawater and measuring its DIC upon equilibration or ii) calculating DIC in air-equilibrated discharge by measuring temperature, salinity, pH, DIC and TA of the freshly carbonated seawater and then modelling its DIC at ambient air p $CO_2$. In each case the proportional difference in DIC before and after air equilibration must be subtracted from the DIC of the carbonated seawater to yield long-term, gross carbon removed. Net carbon removal is obtained by subtracting all

anthropogenic $CO_2$ emissions incurred in the performance of AWL from gross CDR: gross CDR – emissions = net CDR. Emissions includes those associated with carbonate extraction, processing and transportation; energy usage in water pumping and other operating activities; and infrastructure and maintenance of the system

*Economics.* Estimating the possible economics of AWL systems at scale is essential for making informed decisions on future

RD&D. It requires extrapolating/modelling the costs measured or inferred at research scales. It is therefore important to record carbonate purity, energy usage and efficiency, volumetric reaction rates, water pumping requirements, etc so as to better estimate costs and performance at scale.

*Environmental and social impacts.* As in all OAE, the upstream and downstream environmental and social impacts of AWL

must be considered at research and larger scales. In particular the impacts of:

i) Increased carbonate mineral extraction and processing, though keeping in mind that limestone mining and processing has resulted in massive waste piles of carbonate material whose use could actually benefit land reclamation (Langer et al. 2009).

ii) Seawater pumping, screening and carbonation which can impact resident biota (CEC 2005). When possible, it is best to utilize existing seawater pumping (such as for power plant condenser cooling) to avoid additional impacts of new pumping. If

the $CO_2$ used for carbonation is hot (e.g., from exhaust from a biomass energy plant) this will warm the seawater, potentially affecting downstream biota.

iii) the purity of the carbonate minerals used, in particular the presence of any trace constituents that could have environmental consequences downstream (Chapter 4). Likewise, the purity and temperature of the $CO_2$ used must be considered in evaluating downstream impacts.



iv) the societal consequences of AWL activities including those associated with the upstream increased carbonate mineral extraction, processing and transportation, the footprint of the AWL operation on land and any impacts occurring downstream in the ocean (see Chapter 5).

## 3.4 Ocean Liming

### 3.4.1 Technical summary

Ocean liming is the process of adding lime (CaO) or hydrated lime (Ca(OH)$_2$) to the surface ocean (Kheshgi 1995, Renforth et al., 2013), the dissolution of which increases seawater alkalinity (Reaction 6 and 7; Kheshgi 1995). Lime is conventionally manufactured through the calcination of limestone at >800°C (Reaction 8) using fossil fuels and is used in a range of industries including steelmaking, paper manufacturing, construction, and agriculture.

$$CaO_{(s)} + 2CO_{2(aq)} + H_2O \rightarrow Ca^{2+} + 2HCO_3^- \qquad \text{(R6)}$$

$$Ca(OH)_{2(s)} + 2CO_{2(aq)} \rightarrow Ca^{2+} + 2HCO_3^- \qquad \text{(R7)}$$

$$CaCO_{3(s)} \rightarrow CaO_{(s)} + CO_{2(g)} \qquad \text{(R8)}$$

First proposed for OAE (Kheshgi 1995) the CO$_2$ produced from the kiln (from limestone decomposition and fossil fuel use) 485   must be captured and stored for the technology to result in a net- CO$_2$ removal (Fig. 3.11). However, energy requirements (<6 GJ tCO$_2^{-1}$) and the cost ($70 - 160 tCO$_2^{-1}$) of ocean liming are consistent with other engineered CDR approaches (Renforth et al., 2013). Others have shown that the integration of biomass and hydrogen energy vectors may improve the process carbon balance and cost feasibility (Caserini et al., 2019).

Kiln technologies for CaO (often referred to as 'burned lime' or 'quicklime') production are diverse and include upright or inclined shafts, rotating shafts, and parallel or contraflow introduction of fuel and feedstock. The selection of kiln type depends on the product material characteristics, quality of the limestone feedstock, local market demand, fuel type and availability, and finance availability (European Commission 2013).





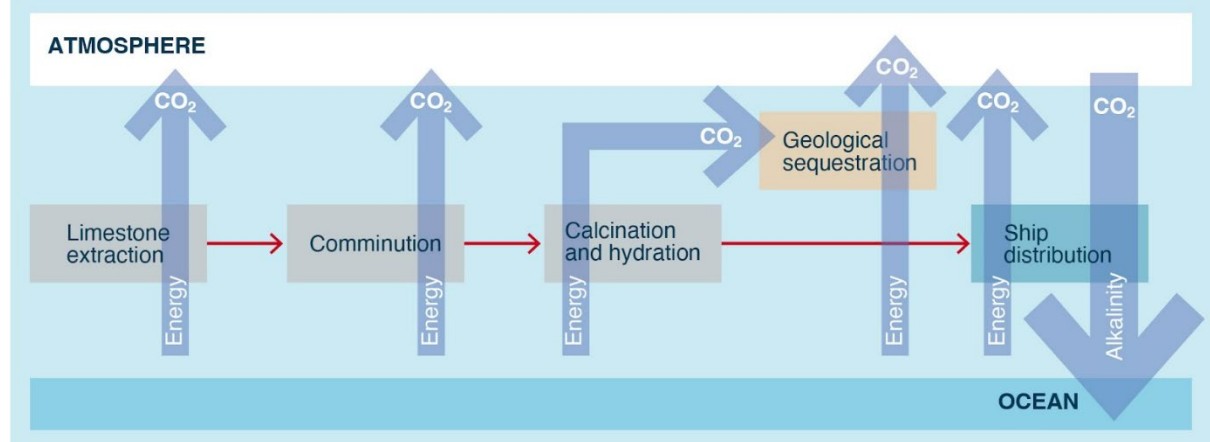


**Figure 11. A simplified process flow diagram of ocean liming (Renforth et al., 2013).**

Limestone decomposes at high temperature by solid-state diffusion of $CO_2$ through the material. The resulting CaO retains the overall volume of the initial calcite but with increased internal porosity (Fischer 1955). As such it is possible to create lump
lime (larger particles of lime produced from similarly sized feedstock limestone).

Powdered $Ca(OH)_2$ is produced by adding a stoichiometric volume of water to CaO ('hydration', if excess water is used this is referred to as 'slaking', Reaction 9). This hydration reaction is exothermic resulting in the breakdown of CaO to a fine powder of $Ca(OH)_2$. This is thought to be via a topochemical mechanism (Gartner 2018) which produces $Ca(OH)_2$ particles
around 2-5 μm (Yakymechko et al 2020), these particles often form larger aggregates of 30-40 μm bound together by weak Van der Waals forces (Yakymechko et al 2020). They are also more porous than CaO resulting in a higher specific surface area (Moropoulou et al., 2001). There is some literature to suggest that the size of the particle may be controlled by the slaking temperature, (e.g., steam slaking Pesce et al., 2023). Furthermore, the fast reaction rates resulting from small particle size and high surface area are useful in most traditional applications. but may not be appropriate for ocean liming (see below)


$$CaO_{(s)} + H_2O \rightarrow Ca(OH)_{2(s)} \qquad (R9)$$





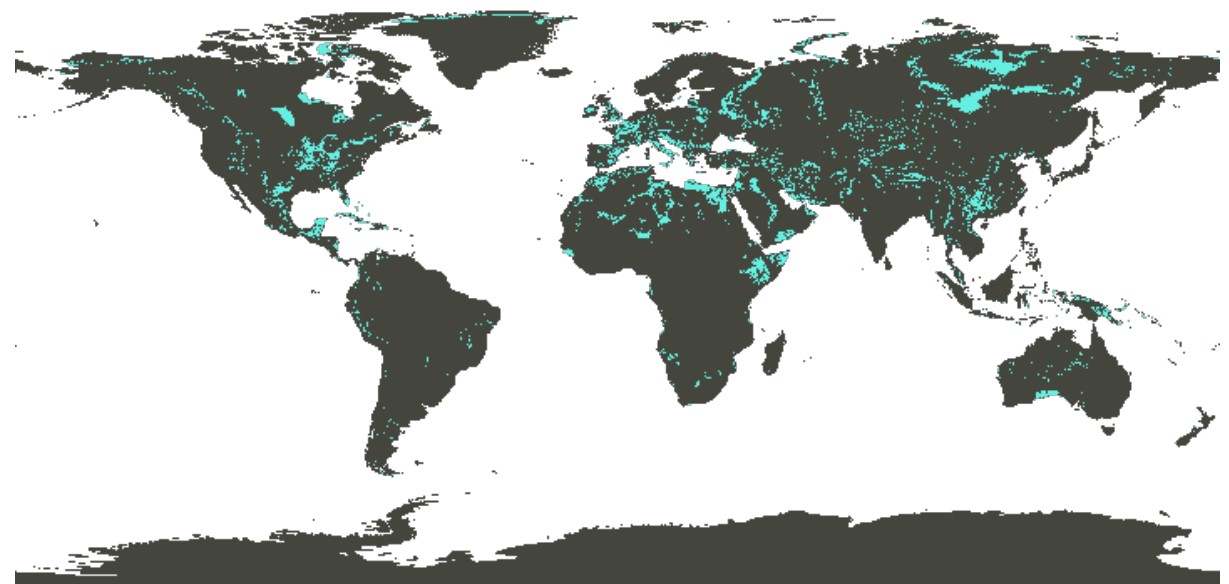

**Figure 12. Global distribution of carbonate sedimentary deposits (data from Hartmann and Moosdorf (2012), and map adapted from Renforth et al., 2022).**


Provided that the CCS is in place to capture the $CO_2$ emissions from limestone decomposition during calcination, OL can be carbon-negative even when current technology and not fully decarbonized energy is used. Specifically, an LCA on OL (Foteinis et al., 2022) revealed that the main environmental hotspots of the process were limestone calcination, where fossil fuel is consumed for heat generation, followed by CCS, which is energy intensive, while mining, hydration, and ocean

spreading affected the environment impact to a much lesser extent. The LCA results were also sensitive to transportation means and distance (although carbonate sedimentary rocks are widely distributed, Fig. 3.12), and particularly to the kiln technology and fuel type during calcination. When best available technology is used, along with renewable electricity to drive the process (e.g., calcination using plasma torches), then OL's environmental performance is optimized. If the low-grade heat generated during hydration is also recovered and used for district heating, then avoided emissions could be also realized, which can be

larger than the process life cycle emissions. In this sense, not only the full amount of CDR is credited to OL but also avoided emissions are achieved (Foteinis et al., 2022).

It is possible to create magnesium oxides and hydroxides, either through the calcination of magnesium carbonates (McQueen et al 2020) or through extraction from magnesium silicates (Renforth and Kruger 2013). While the calcination temperature and

energy of magnesite is substantially lower than that of calcite, global reserves are approximately ~10 Gt (McQueen et al 2020), which suggests their exploitation at scale for ocean liming may be limited. Renforth and Kruger explore the coupling of mineral carbonation and ocean liming in which Mg is extracted from abundant silicate minerals through carbonation and then calcined to produce magnesium oxide/hydroxide materials (Fig. 3.13).



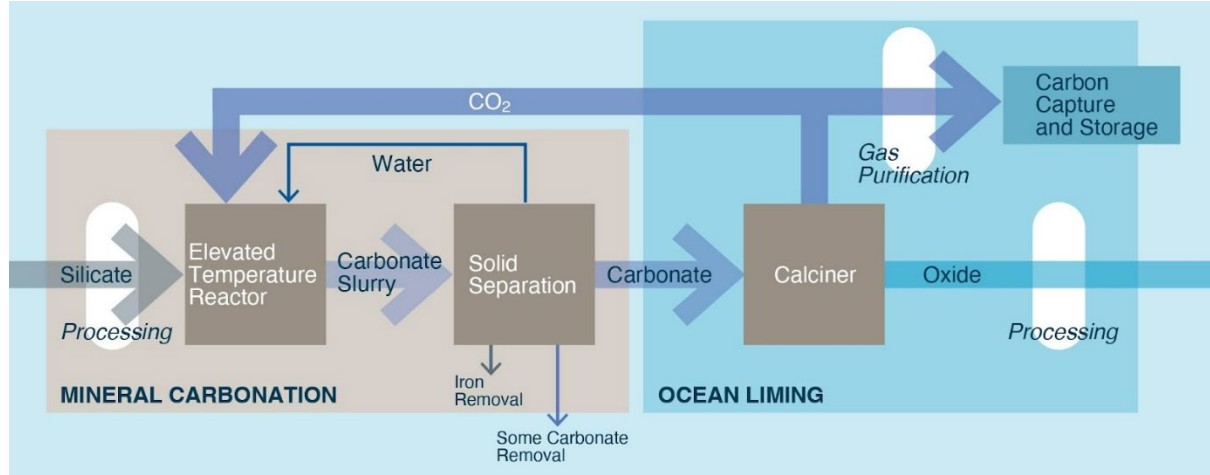


**Figure 13. A coupled mineral carbonatation ocean liming system proposed by Renforth and Kruger 2013.**

Processes have been suggested for the extraction of Mg from silicate minerals and the creation of $Mg(OH)_2$ for the purpose of capturing carbon dioxide from flue gasses (Madeddu et al., 2015, Nduagu et al.,2012), but may be suitable for OAE as well.

Nduagu et al., (2012) suggest a multistep process in which serpentinite (a Mg rich rock) is heated at 400°C with solid ammonium sulphate creating a solid magnesium sulphate and silica, and evolving ammonia and water as gas. The silica can be leached by washing the solid product with water, following which bringing the ammonia gas back into contact with the Mg sulphate creates high pH conditions in which $Mg(OH)_2$ can precipitate. An alternative approach, (Madeddu et al., 2015) heats solids of NaOH and an olivine rich rock at 180°C, forming $Mg(OH)_2$ and a Na silicate. Both approaches propose to use the

$Mg(OH)_2$ for direct reaction with flue gas, and no work has explored their potential for OAE.

### 3.4.2 Considerations for best research practices

*Lime/hydrated lime production:* Lime can be easily created for small scale laboratory experiments by calcining limestone (or laboratory grade calcium carbonate) in a furnace at 900°C. It is possible to sinter lime at temperatures >1,100°C, which would

result in a lower reactivity. While sintering is often undesirable for commercial lime, the effect may be useful for OAE to reduce particle dissolution rate and prevent oversaturation of carbonate minerals in seawater. Lime products can also be sourced from commercial suppliers. The reaction of burnt lime (CaO) with water is highly exothermic and its fire-safety risk should be considered when storing or using in the laboratory.

*Carbonation prior to experimentation:* Lime and hydrated lime readily react with atmospheric $CO_2$ so should be produced or sourced as near to the start of the experimental work as possible. The materials can be stored in airtight and/or desiccated





containers to minimize carbonation. However, it is difficult (if not impossible) to limit carbonation and will certainly be present within commercially sourced material. Carbonate content should be measured (e.g., through mass loss on ignition) before the experiment.


*Reactivity, spontaneous precipitation in seawater:* Commercially sourced hydrated lime has been manufactured for applications in which high reactivity is desirable. Hartmann et al, (2023) have shown that additions of these particles (0.7 mg $Ca(OH)_2$: 1 g seawater) may result in spontaneous precipitation of carbonate minerals. Caserini et al., (2022) modelled an initial particle density of approximately 80 mg $Ca(OH)_2$: 1 g seawater, diluting to <0.6 mg $Ca(OH)_2$ : 1 g seawater (assuming

a 10 kg s$^{-1}$ addition through a single discharge nozzle in the wake of a ship) within about 30 seconds. Experiments that add hydrated lime to solution should use an initial concentration <0.7 mg $Ca(OH)_2$ assessing particle dissolution across as $10^2$ - $10^4$/min range of dilution.

## 3.5 Hydrated carbonate mineral formation

### 3.5.1 Technical Summary

There are several hydrated carbonate minerals such as ikaite, monohydrocalcite, nesquehonite, hydromagnesite, and amorphous calcium carbonate that are undersaturated in the surface ocean and are thermodynamically likely to dissolve and increase alkalinity when added to seawater (Table 3.1). The occurrence of these minerals and a method for their industrial production is presented below,


Ikaite ($CaCO_3 \cdot 6H_2O$) precipitates from aqueous solutions close to freezing conditions (Boch et al., 2015) in a narrow temperature range below ca. 4 – 8 °C and depending on ionic strength down to negative temperature values (e.g., in highly saline solutions down to −8 °C; Hu et al., 2014, Papadimitriou et al., 2014). Alternatively, elevated pressure conditions (>3 kbar at 25 °C) facilitate the crystallization of ikaite (Marland, 1975, Shahar et al., 2005). The solubility of ikaite is higher

compared to the anhydrous calcium carbonate polymorphs calcite, aragonite and vaterite (Brecevic and Nielsen, 1993, Marion, 2001). Dissolved compounds such as magnesium, phosphate, sulphate, and organic molecules that inhibit the formation of anhydrous calcium carbonates, favour the nucleation of ikaite (Brooks et al., 1950, Bischoff et al., 1993b). Outside these restricted environments ikaite dehydrates and disintegrates rapidly, (within minutes to weeks), preferentially into more stable carbonate minerals and water (Mikkelsen et al., 1999). In some cases, calcite pseudomorphs after ikaite might persist

("glendonite", Greinert and Derkachev, 2004).

Monohydrocalcite (MHC: $CaCO_3 \cdot H_2O$) is a rare mineral in geological settings (Nishiyama et al., 2013), but is frequently associated with other calcium and magnesium carbonate minerals, such as calcite, aragonite, lansfordite, and nesquehonite





(Nishiyama et al., 2013). Monohydrocalcite has been observed in air conditioning systems (Nishiyama et al., 2013), in
'moonmilk' deposits in caves (Nishiyama et al., 2013), and in beach sands formed from algal spume (Nishiyama et al., 2013).
It has been reported as a significant component of the decomposition of ikaite in the towers of the Ikka Fjord, West Greenland
(Nishiyama et al., 2013). Both laboratory studies and natural observations have indicated that the formation of MHC requires
the presence of magnesium in the solution (Nishiyama et al., 2013), and possibly forming via an Mg-rich amorphous precursor
(Nishiyama et al., 2013).


The magnesium carbonate mineral nesquehonite ($MgCO_3 \cdot 3H_2O$) precipitates at room temperature from supersaturated
solutions rich in magnesium and bicarbonate (Hopkinson et al., 2008). It is metastable and transforms into hydromagnesite in
ambient conditions (e.g., Kazakov et al. 1959), which may be responsible for some naturally occurring hydromagnesite (Davies
and Bubela 1973). Mafic and ultramafic mining wastes, by virtue of their high calcium and magnesium content, are prone to
forming numerous carbonate species upon contact with atmospheric $CO_2$ depending on the environmental conditions that
prevail at the stockpiles, but metastable nesquehonite was reported to be the dominant magnesium carbonate forming in
ambient conditions (Zarandi et al., 2017). Upon rising the temperature above 50 °C, nesquehonite evolves into
thermodynamically more stable products with lower $CO_2$:Mg ratios (Zarandi et al., 2017).

Hydromagnesite ($Mg_5(CO_3)_4(OH)_2 \cdot 5H_2O$) is a naturally occurring hydrated magnesium carbonate (e.g., Königsberger et al.
1999). At the Woodsreef Asbestos Mine, New South Wales, Australia, weathering of ultramafic mine waste sequesters
significant amounts of $CO_2$ in hydromagnesite ($Mg_5(CO_3)_4(OH)_2 \cdot 4H_2O$) (Oskierski et al., 2021). Mineralisation of $CO_2$ in
above-ground, sub-aerially stored tailings is driven by the infiltration of rainwater dissolving Mg from bedrock minerals
present in the tailings (Oskierski et al., 2021). Complete dissolution of source minerals, or formation of Mg-poor products
during weathering, is expected to transfer Mg into solution without significant alteration of the Mg isotopic composition
(Oskierski et al., 2021). The main mineral sources of Mg in the tailings (silicate, oxide/hydroxide and carbonate minerals) are
isotopically distinct and the Mg isotopic composition of fluids and thus of the precipitating hydromagnesite reflects both
isotopic composition of source minerals and precipitation of Mg-rich secondary phases (Oskierski et al., 2021). The consistent
enrichment and depletion of [26]Mg in secondary silicates and carbonates, respectively, underpins the use of the presented
hydromagnesite and fluid Mg isotopic compositions as a tracer of Mg sources and pathways during $CO_2$ mineralisation in
ultramafic rocks (Oskierski et al., 2021).

Amorphous calcium carbonate (ACC) is unstable under normal conditions and is found naturally in taxa as wide-ranging as
sea urchins, corals, molluscs, and foraminifera. It is usually found as a monohydrate, holding the chemical formula
$CaCO_3 \cdot H_2O$, however, it can also exist in a dehydrated state, $CaCO_3$ (Rodriguez-Blanco et al., 2011). ACC has been reported
for over 100 years when a non-diffraction pattern of calcium carbonate was discovered by Sturcke Herman, exhibiting its
poorly ordered nature (Rodriguez-Blanco et al., 2010). The structure and chemistry of ACC is complex with several forms of



ACC classified according to their water content, local order, and mode of formation (e.g., abiotic vs biogenic). A key variable is the amount of structural water. Hydrated-ACC can contain up to ~1.6 mol of water per mole of $CaCO_3$, yet several less
hydrated and even anhydrous forms of ACC have been described (Rodriguez-Blanco et al., 2010; Bots et al., 2012).

**Table 1 A summary of potential hydrated carbonate minerals for ocean alkalinity enhancement (see Renforth et al., 2022).**

| Hydrated carbonate mineral | Chemical formula | Reported occurrence | Gibbs free energy of hydration reaction from unhydrated forms[a] (kJ mol[-1]) | Gibbs free energy of dissolution reaction in seawater[a] (410 ppm @ 25°C) (kJ mol[-1]) |
|---|---|---|---|---|
| Monohydocalcite | $CaCO_3$ $H_2O$ | Formation of MHC requires magnesium in the solution in spite of the incompatibility of magnesium into the MHC structure. Monohydrocalcite has been observed in air conditioning systems, and in moonmilk deposits in caves, both probably formed from spray of carbonate rich fluids. | 4.0 | -2.4 |
| Ikaite | $CaCO_3$ $6H_2O$ | Naturally occurring, metastable hydrated calcium carbonate mineral that forms in cold (<15C), alkaline, nutrient-rich waters. | 10.2 | -8.9 |
| Nesquehonite | $MgCO_3$ $3H_2O$ | The magnesium trihydrate carbonate nesquehonite readily precipitates from solutions of magnesium bicarbonate at room temperature. | 17.1 | -11.5 |
| Hydromagnesite | $Mg_5(CO_3)_4$ $(OH)_2$ $5H_2O$ | Hydromagnesite is an abundant naturally occurring magnesium hydroxyl carbonate (e.g., Königsberger et al. 1999; Russell et al. 1999; Edwards et al. 2005b) that constitutes a large and potentially reactive sink for C. | 16.9 | -28.9 |





| | | | | |
|---|---|---|---|---|
| Hydrated amorphous calcium carbonate | $CaCO_3$ $xH_2O$ | ACC is unstable under normal conditions and is found naturally in taxa as wide-ranging as sea urchins, corals, molluscs, and foraminifera. | - | - |
| Calcite | $CaCO_3$ | Naturally abundant in limestone | - | 1.6 |
| Magnesite | $MgCO_3$ | Accessory mineral in limestone, alteration product in weathering of ultrabasic rock | - | 5.7 |

a. Negative values denote an exothermic/spontaneous reaction.

Hydrated carbonate minerals are relatively rare and thus insufficient to meet the demand of a scaled up OAE industry. As such, they would need to be created. Renforth et al., (2022) suggest a process by which limestone is dissolved in water in an elevated $CO_2$ pressure reactor (~2 bar). The water is then degassed in lower pressure reactors (~20 mbar) under vacuum to evolve/recycle the gaseous $CO_2$ and create conditions in which carbonate minerals are likely to precipitate from the solution (Fig. 3.14). If the precipitation environment is cooled or in the presence of calcite precipitation inhibitors, then Renforth et al., (2022) suggest that ikaite will form.



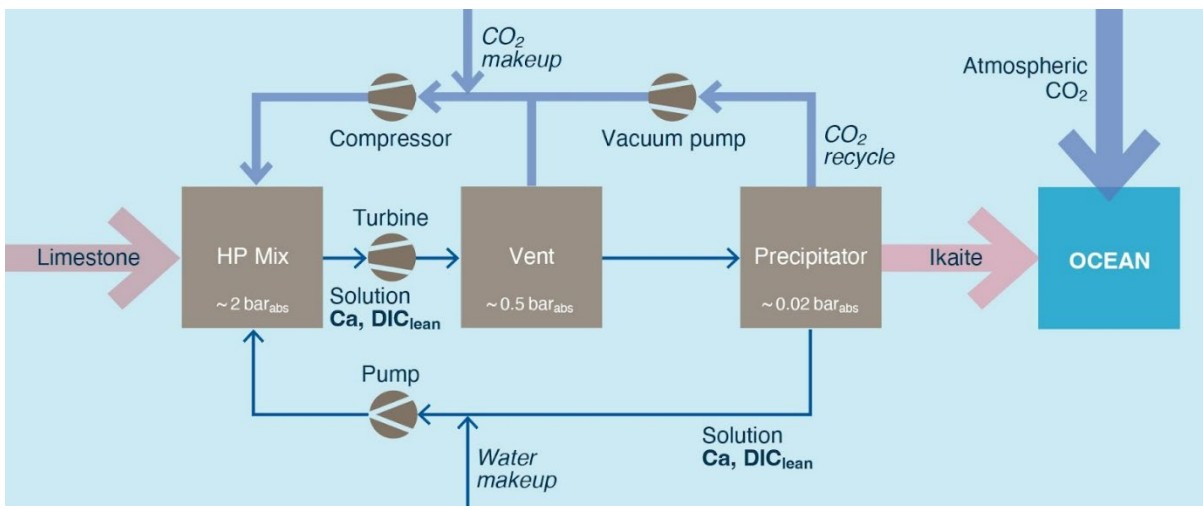


**Figure 14 A CO$_2$ pressure swing process for creating a hydrated carbonate (adapted from Renforth et al., 2022).**

There are several benefits for considering hydrated carbonate mineral addition for OAE. The chemical energy imparted into lime or hydrated lime during production is released during its dissolution and carbonation in the surface ocean. This energy is impossible to recover. The production of hydrated carbonates requires considerably less energy.

### 3.5.2 Considerations for best research practices

*Summary of method of precipitating hydrated carbonates:* The multi-step system pressure system for its mass production (Fig. 3.14) may not be convenient or necessary for all laboratory studies that may want to consider the impact of ikaite dissolution 645 on seawater biogeochemistry. Existing methods for precipitating hydrated carbonates involve the reaction of an alkaline liquid (usually NaOH or Na$_2$CO$_3$) with calcium or magnesium chloride (e.g., Lennie et al. 2004). It is not yet clear if characteristics of these materials differ from those that might be produced from a pressure swing system.

*Stability of hydrated carbonates:* The feasibility of using hydrated minerals for OAE requires that the produced mineral 650 remains stable for sufficient time to be added to the ocean and dissolve. Similarly, experiments performed using hydrated minerals should have sufficient confidence that the materials added to seawater have not converted to more stable anhydrous polymorphs. If these minerals were to transform into calcite or magnesite before addition it could result in a reduction of alkalinity by seeding additional carbonate precipitation. Experimental work suggests that synthetic ikaite can be stable for days at low temperature and that it increases alkalinity when dissolved in seawater (Renforth et al., 2022).




_Methods of detection:_ Hydrated carbonate minerals are identifiable by several techniques including Raman spectroscopy, Fourier Transform Infrared (FTIR) spectroscopy, X-Ray Diffraction (XRD), Electron Probe X-ray Microanalysis (EPMA) and Scanning Electron Microscopy (SEM).

Raman spectroscopy is a non-destructive method that requires little or no sample preparation. Moreover, all the bands related to carbonates in the mid-infrared region have characteristic positions which make it easy to differentiate from other minerals. Raman analyses of carbonates have long been used in mineralogical and geochemical research (see Kim et al., 2021 and references therein).

FTIR is rapid (a few minutes per sample), does not require hazardous chemicals, has a small sample requirement (~ 1 mg), and produces several distinguishable carbonate bands in its spectrum. Diffuse reflectance infrared Fourier transform spectroscopy (DRIFTS) is a form of FTIR with additional advantages compared to transmission FTIR. DRIFTS does not require sample dilution in infrared transparent material, thus reducing sample preparation time, its sample-holding microcells allow for fixed volumes, and the sample is recoverable after analysis. DRIFTS is a method that has been used for identifying

and quantifying calcite and dolomite in natural sediments. Few studies have used spectroscopic techniques to quantify carbonate in non-carbonate geological matrices (Bruckman and Wriessnig 2013; Du et al. 2013; Tatzber et al. 2007).

XRD is the most commonly used tool for identification of major minerals. In addition to qualitative analysis, quantitative XRD is possible because the peak intensities of a given mineral in the diffractogram are proportional to the weight percent of that

particular mineral in the sample. However, peak intensities are also a function of the mineral's absorption coefficient, particle size, degree of crystallinity and the preferred orientation of the sample, this means that compared to qualitative analysis quantitative XRD requires more specialized expertise to produce accurate results.

## 3.6 Mineral addition to pelagic coastal environments

### 3.6.1 Technical Summary

Using the ocean's surface waters for mineral addition to increase ocean alkalinity has been the focus of interest of numerous research projects (Renforth & Henderson, 2017), starting with exploring suitable minerals to achieve increased alkalinity while investigating potential side effects on marine life and ecosystems. An obvious problem for the community of researchers involved in OAE has been identifying and generating the particle type and size required to allow for dissolution, classically

considered to be smaller than 63 μm (Hangx & Spiers, 2009). However, while this particle size is comparably easy and energy efficient to produce, residence times in the water column would for most minerals be too short to allow for dissolution and particles would sink out too rapidly. To avoid mineral loss via sinking, particle sizes of <1μm would be required (Köhler et



al., 2013), which are difficult to produce in a climate-neutral manner, and their application might be harmful for humans leading to respiratory problems (e.g., Doelman et al., 1990). One way to work with bigger grain sizes is to turn to coastal
systems, where particles would sink to the seafloor and be transported back into the water column by natural turbulence, allowing for increased dissolution over time.

Classically, silica minerals such as olivine (forsterite or fayalite) and carbonate sedimentary rock such as limestone have been suggested to increase alkalinity in an efficient way (Renforth & Henderson, 2017). However, large scale experiments are still
a rarity (see also Chapters 4.4 and 4.5). Other possible minerals for this purpose are basalt or serpentine. When choosing minerals, an ambition should be to ensure availability near the application location to reduce the carbon footprint, thus limiting the choice of mineral and again emphasizing the advantage of coastal applications as compared to open ocean applications.

Another concern when applying alkaline minerals is the stability of alkalinity due to possible formation of carbonate phases.
This results from the ocean's supersaturation in calcite and aragonite (Sarmiento & Gruber, 2006). If an increase in alkalinity is introduced along with an increase in carbonate ion concentrations, the supersaturation would increase even more, which has been suggested to lead to solid carbonate precipitation (Fuhr et al., 2021, Moras et al., 2022; Hartmann et al., 2023). This, in turn, would decrease surface alkalinity causing an effect opposite to the desired one. One proposed solution to address this challenge is the application of $CO_2$-equilibrated alkaline solutions to minimize the risk of losing alkalinity due to carbonate
phase formation.

Open-ocean and water-column silicate mineral applications have the potential to increase both the chemical and the biological carbon pump. Here, the biological carbon pump was of interest due to its potential to remove $CO_2$ on a timescale of several thousands of years (Longhurst et al., 1995, Petit et al., 1999, McNeil et al., 2003). The line of reasoning was often based on
Earth's historical considerations, with cold periods in Earth history having been related to increased photosynthetic activity by phytoplankton (Kirschvink, 1992, Penman & Rooney, 2019). However, recent studies suggest that mineral additions can, besides benefits, also pose risks to marine life, including primary producers (Glass & Dupont, 2017, Bach et al., 2019). This effect is likely related to the increased concentrations of trace metals enriched in the minerals of choice. Earlier dissolution experiments with olivine have shown that an increase of alkalinity of about 100 µmol $L^{-1}$ led to an increase of ~3 µmol Ni $L^{-1}$
due to non-stoichiometric dissolution of the heterogeneous material, this equals to approximately 3 times the natural concentration in seawater (Montserrat et al., 2017), within the toxic range for many eukaryotic microalgae (Glass & Dupont, 2017). Other trace metals present in alkaline minerals, including Cu, Cd, Cr, or other heavy metals, (Simkin & Smith, 1970, Beerling, 2017) have less clearly understood effects. Anthropogenic materials including from mining or cement production could also contain a variety of trace metals at concentrations yet to be determined, which might become particularly
problematic to organisms of higher trophic levels in which they accumulate (Garai et al., 2021).



It is essential to consider whether the added minerals remain in the water column and impact the growth of the vital primary producers in the food web. Another important consideration is whether adding alkaline minerals impacts the local communities in the treated area and leads to any changes. Moreover, a possible change in the local communities might see the appearance of organisms that release other greenhouse gasses, potentially offsetting the sequestration achieved through the treatment.

Careful considerations have to be undertaken in order to ensure safe-minimum standards. One important aspect is that field applications cannot exceed environmental quality standards (EQS, (European Commission, 2017)). EQS defines the threshold concentration of potentially harmful toxic metals, like nickel and chromium. The impact of minerals enriched in those trace metals on biodiversity and ecosystem function is difficult to study, as minerals are not necessarily homogeneous and contain similar trace metal concentrations. Trace metal concentrations, however, may limit the amount of mineral addition, especially of olivine-rich rocks that can be deployed in marine habitats (e.g., Flipkens et al. (2021)). The potential difficulty arising from introducing toxic compounds is likely less pronounced in open ocean environments due to export and rapid dilution. Coastal environments can\ have limitations, especially with regard to sediments, which already have a (natural) background in trace metals and accumulate trace metals over time (see chapter 3.6.2). Furthermore, seawater pH should be kept within a natural seawater range (pH<9; Pedersen and Hansen, 2003). Even though phytoplankton can be adapted to a wide range of pH, the growth rates of a majority of species can be influenced significantly (Hinga, 2002, Penman & Rooney, 2019). In addition, a pH > 9.5 can promote the precipitation of aragonite, as well as brucite together with phosphorus and silicate (Hartmann et al., 2023). On the other hand, depending on the deployed material, specific plankton groups might benefit from, for example, iron additions (Boyd & al., 2000).

To scrutinize the effect of OAE as a tool to mitigate climate change, it is necessary to also investigate the impact of mineral addition, pH and alkalinity changes on non- $CO_2$ greenhouse gas production. Two important greenhouse gasses are methane and nitrous oxide, with warming potentials of approximately 70 and 300 times that of $CO_2$, respectively (Bange, 2006). While the ocean is a minor source of methane to the atmosphere, it contributes about one third of nitrous oxide emissions to the atmosphere (Bange, 2006), making it critical to understand any potential impacts of OAE on its formation. $N_2O$ is chiefly produced biologically, with the microbes producing it known to be sensitive to changes in pH (Thomsen et al., 1994, Seeländer, 2023), however, so far, the few available studies indicate a reduction in $N_2O$ production if pH and alkalinity are increased.

When considering the technical challenges within the pelagic environment for OAE, one of the primary considerations is how to effectively measure the $CO_2$ offsets generated from techniques that utilize the pelagic environment, therefore a standardization of MRV (measurement, reporting, and verification) is a necessity to secure procedures that ensure the accuracy and precision of measurements (see also chapter 6). Within that, a reliable and well-established analysis technique for determining relevant parameters (e.g., alkalinity) is required to ensure the accuracy and comparability of data. Ecosystem diversity in pelagic environments will vary according to the location because of the biological, chemical and physical





parameters, due to for example ocean currents, temperatures, wind etc. Therefore, considerations from many disciplines are essential to explore the complexity.

To carry out coastal pelagic OAE research an understanding of benthic-pelagic coupling is required to assess the impact on benthic systems and ecosystems, and to be able to maximize the use of this coupling for OAE. Further, it is critical to assess the local and regional biodiversity by means of meta-omics, flow cytometry or similar high-resolution methods for microbial life to then understand what thresholds for trace metal additions would introduce toxic effects as defined in Bach et al. (2019) and references therein. For macro-lifeforms, targeted ecotoxicological assessments are required to avoid damage to, and heavy metal accumulation in, top-predators, and to establish robust thresholds.


### 3.6.2 Considerations for best research practices

Application of particles is challenging, (see also 3.7.2), as small sizes are required to assure water column dissolution, but larger particles (~63 μm) can be used if the goal is to achieve OAE in both the benthic and pelagic parts of a coastal system, where resuspension of particles can be useful to achieve mixing. However, any particle addition might lead to shadowing thus
impacting photosynthetic organisms in the pelagic and benthic realm, requiring a thorough understanding of both, benthic and pelagic primary production, to avoid harming the basis of the ecosystem. In addition, mechanical stress can be imposed on benthic organisms by adding particles, data, however, is limited, here, and will have to be gathered for every system, individually.

While those effects will vary largely for each ecosystem, trace metal toxicity can be avoided. Before particles are added, we recommend a thorough analysis of the major elements in the mineral of choice to quantify the effect of mineral addition on the total alkalinity change potentially obtained in the system, but also to define the upper limit of additions with respect to trace metal toxicity. For some organisms, including pelagic primary producers, thresholds are available from the literature, however, to understand the impact of trace metal toxicity on complex food webs testing before application should be carried out in
mesocosms and benthocosms containing assemblages or subsets of the natural communities present in the ecosystem of choice. At a minimum, key species should be tested individually for their trace metal (and pH) tolerance to avoid damage to biodiversity.

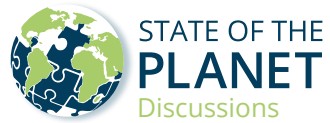

### 3.7 Addition to the coastal seafloor

### 3.7.1 Technical summary

Continental shelves comprise only about 7% of the global surface ocean, yet account for up to 30% of the oceanic primary production (Gattuso et al., 1998) and between 10 and 25% of the present-day $CO_2$ uptake (Regnier et al., 2013). Shelves are also important areas for cation and TA turnover related to detrital mineral dissolution and authigenic carbonate and clay precipitation (e.g., Linke et al., 1994; Jeandel et al., 2011; Jeandel & Oelkers, 2015; Torres et al., 2020). Assuming an ice-free
surface area of continental shelf seas of $22 \times 10^6$ km$^2$, natural carbon uptake in coastal waters has been estimated to be -0.19 Pg Cyr$^{-1}$, showing the importance of shelves as a natural global sink of atmospheric $CO_2$ (Laruelle et al., 2014, 2018).

The addition of ground minerals to the shelf seafloor may enhance this uptake even further, and as such are candidate locations for OAE. Large-scale mineral application would be logistically convenient, either using excavators directly on the beach or from small vessels or barges in offshore shallow waters. Several companies and initiatives already take advantage of this
relatively easy CDR-implementation and study the effectiveness of mineral dissolution in the field.

The most obvious advantage of adding particulate minerals to the seafloor compared to the water column is the required grain size. Compared to water column deployment, for which the required grain sizes are <1µm in order to avoid rapid sinking (Hauck et al., 2016), the optimum grain sizes needed for seafloor deployment are proposed to range between 0.2 and 1.4mm
(Schuiling & de Boer, 2011, Strefler et al., 2018), which is economically attractive. However, these recommendations lack thorough experimental testing, both in the laboratory and in the field.

Further, the choice of grain size depends on the region of deployment. Two potential coastal regions for mineral addition are low- and high-energy environments (modified after Meysman & Montserrat, 2017) (Fig. 3.15). High-energy environments are
characterized by extensive water mass movement, such as surf zones, thereby providing a natural grinding mechanism allowing larger, less costly, grain sizes to be used (Meysman & Montserrat, 2017). In contrast, low-energy environments require the addition of smaller grain sizes (~20 to 100µm) as a higher surface area maximizes mineral dissolution (Oelkers et al., 2018). In this environment, mineral dissolution is enhanced by biota through bioturbation and microbial metabolism (Meysman & Montserrat, 2017). These two distinct environments are discussed below with respect to their advantages and disadvantages.





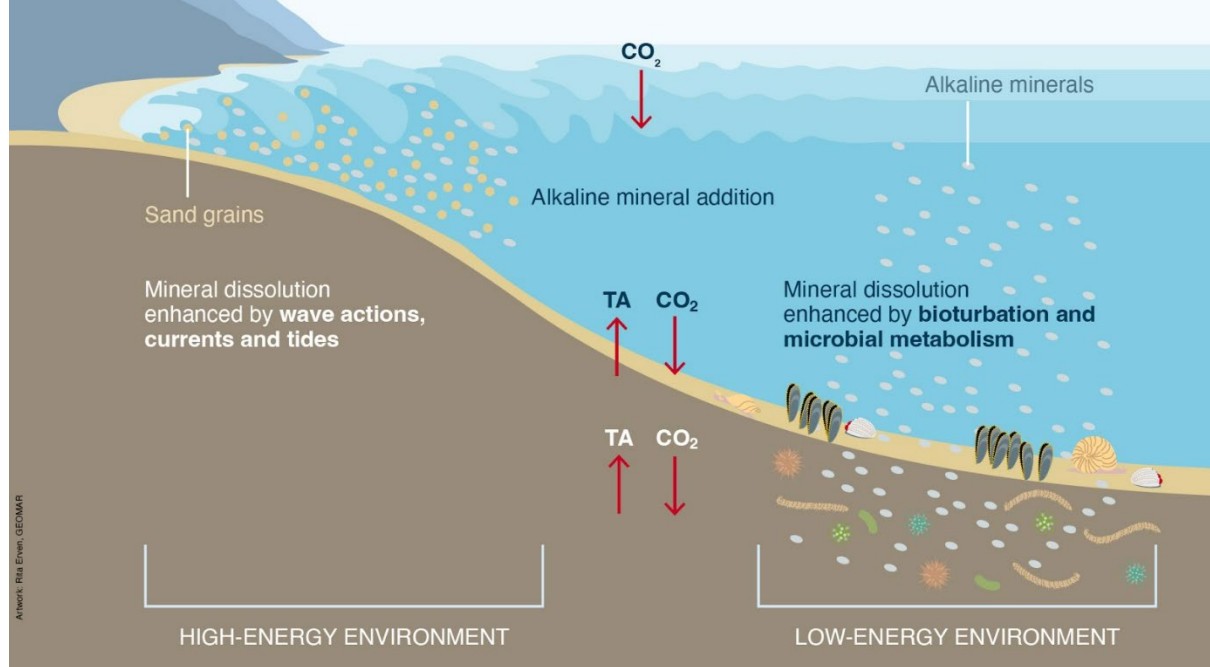


**Figure 15 High- and low-energy environments with different regional advantages favouring mineral dissolution.**

*High-energy environments:* These environments are located along the coastal surf zone or the shallow shelf, where wave action, currents and tidal activity lead to natural mineral erosion. The required grain sizes can be relatively large (mm-scale)

for this environment, given that the constant movement naturally grinds the minerals, continuously exposing fresh reactive mineral surfaces and enhancing dissolution. Given that alkalinity enhancement directly occurs close to the air-sea interface in these well-mixed shallow water settings, the impact on atmospheric $CO_2$-uptake is immediate.

Detection of mineral dissolution and alkalinity production in these high-energy environments is challenging due to rapid

mixing, dilution and dispersal of solutes. Minerals may also be transported along- and offshore away from their deployment location. This requires carefully designed field experiments to assess the efficiency of this OAE approach with regard to $CO_2$ draw-down (see section 3.6.2.2 and chapter 4.5). Furthermore, these factors hamper accurate evaluation concerning MRV for commercial carbon removal purchases (see Chapter 6). An advantage of mineral dissolution in coastal surface sediments is the lower risk of secondary mineral formation. In an open environment, supersaturation levels potentially triggering carbonate or

phyllosilicate precipitation in conjunction with other relevant factors (e.g., pH, DIC, TA) are more unlikely to be attained. $CO_2$-consumption may therefore be more efficient compared to low-energy environments. See Table 3.2 for a summary of advantages and disadvantages for mineral deployment in high-energy environments.





*Low-energy environments:* Shelf environments comprise the waters below the wave base down to 200m that are not
significantly affected by normal wave action. The continental slope and deep seafloor are not considered here given that the
exchange of bottom waters with surface ocean waters may be too slow to be relevant for near-term $CO_2$-reduction strategies
(Smith et al., 2023).

The shelf sediment is mostly sand and mud (Schulz & Zabel, 2006), providing an ideal environment for mineral dissolution
by taking advantage of the 'benthic weathering engine' (Meysman & Montserrat, 2017). Here, two mechanisms can be
distinguished that potentially accelerate mineral dissolution, that is, microbial metabolism and bioturbation by macro-
organisms.

The degradation of organic matter by microbial activity in fine-grained sediment creates a unique microenvironment that may
be conducive to dissolution of some types of minerals. The microbial degradation of organic matter by aerobic and, in
particular, anaerobic remineralization pathways (e.g., denitrification, iron- and manganese reduction and sulphate reduction)
leads to an accumulation of reduced forms of S, N, Fe and Mn in both the dissolved and particulate phases. If these compounds
are exposed to oxygen or nitrate at the sediment surface they can be oxidized rapidly, leading to a local decrease in pH to <6.
Under these conditions, carbonate dissolution may take place if porewaters become undersaturated with respect to the
dissolving mineral phase (Jahnke & Jahnke, 2000). Silicate dissolution strongly depends on the mineral type, whereby Mg-
silicate dissolution (e.g., olivine) is enhanced at low pH and aluminosilicates at high pH (Oelkers et al. (2018) and references
therein). Mineral dissolution may be facilitated or even enhanced through the action of electrogenic cable bacteria that can
oxidize S by shuttling electrons from subsurface anaerobic sediments to the oxic surface layer (Meysman et al., 2019).
However, cable bacteria are not thought to be highly active in bioturbated sediments, such that their potential impact on
alkalinity enhancement by mineral dissolution may not be quantitatively significant at the regional scale.

The effect of burrowing by large macroorganisms, known as bioturbation, enhances mineral incorporation into the sediment
matrix and, consequently, bringing the minerals in contact with the acidic pore fluids and enhancing benthic-pelagic exchange
(Neumann et al., 2021). Additionally, the digestive systems of macroorganisms, with their high enzymatic activity, low pH,
mechanical abrasion, and digestion has been shown to increase silicate and carbonate dissolution (Cadée, 1976; Volkenborn
et al., 2009). However, this process is poorly understood and its significance for mineral dissolution is unknown at regional
scales.

A large drawback of the low-energy benthic environment for OAE is the high probability of secondary mineral precipitation.
Formation of authigenic carbonates or phyllosilicates releases $CO_2$ (e.g., Wallmann et al., 2008; Torres et al., 2020) and
directly counteracts the envisioned TA release and $CO_2$ uptake by mineral addition. Authigenic aluminosilicate formation was
recently found to be a large Si sink in the global marine Si cycle, releasing $CO_2$ and consuming TA (e.g., Wallmann et al.,





2008; Rahman et al., 2017; Tréguer et al., 2021) on time scales of weeks to months, that is, much faster than previously considered ($10^3$ years) and impacting element cycles on human timescales (Geilert et al., 2023). Authigenic, inorganic

carbonate precipitation at the sediment-water interface or within the sediment column is triggered by alkalinity production from anaerobic microbially-mediated reactions (Sun & Turchyn, 2014). In the context of mineral addition to increase ocean alkalinity, these secondary mineral formations may play a major role in the net $CO_2$-sequestration efficiency, as recently shown in laboratory experimental studies (Fuhr et al., 2022; Moras et al., 2022; Hartmann et al., 2023). Further research is required to identify the probability and impact of secondary mineral formation on net $CO_2$-turnover with respect to OAE

(Table 3.2 and Chapter 3.7.2). The results from laboratory studies though (Fuhr et al., 2022; Moras et al., 2022; Hartmann et al., 2023) are still debatable with regards to their transferability to the open ocean where secondary mineral saturation states are reached less easily. Mesocosm studies might offer a solution here (Chapter 4.4), in which open ocean conditions can be simulated more realistically and the triggering factors for secondary mineral formation be identified (see also Chapter 3.7.2).

As in the high-energy shallow environments, detection of mineral addition in deeper and fine-grained coastal waters ($50 - 200$ m) can be difficult. The deployment of autonomous instruments on the seafloor, such as benthic chambers, can be used to measure fluxes of alkalinity and other dissolved compounds to/from the seafloor. Depending on the nature of the sediment and the carbon degradation rates, benthic chambers can typically detect $O_2$ consumption and nutrient release on timescales of 1-2 days of continuous deployment in shelf environments (Sommer et al., 2016). However, the attribution of mineral dissolution

to changes in alkalinity is challenging, particularly against the large seawater alkalinity background. Only in the most reactive coastal settings such as upwelling areas can alkalinity fluxes be determined accurately (Ilyana et al., 2013; Dale et al., 2015). The development and incorporation of high-precision pH and p$CO_2$ sensors may provide a solution to detecting small changes in $CO_2$ and alkalinity fluxes due to mineral dissolution. To our knowledge, the suitability of benthic chambers with respect to detection of OAE at the seafloor still requires field testing.


For both high- and low-energy environments, the risk exists that the alkaline minerals will not remain at the site of deployment, either due to seabed erosion by wave action and currents, transport or by burial. In areas where there are strong bottom currents, fine-grained minerals can be eroded and transported. These minerals are then ultimately delivered to regional depocenters in deeper basins in shallow coastal seas such as the Baltic Sea (e.g., Wallmann et al., 2022) or to the continental slopes on open

margins (e.g., Anderson et al., 1994). Once deposited in deep waters, they are removed from the shallow regions where the benthic-pelagic water mass exchange is rapid.

A final point for consideration is the sedimentation rate. On the one hand, high POC sedimentation rates are desirable to guarantee high rates of organic matter degradation and low pH porewaters necessary for enhanced mineral dissolution. On the

other hand, too high sedimentation rates of detrital minerals may be counterproductive, leading to rapid burial of OAE-minerals



below the dissolution zone. These factors need to be factored into the cost-benefit analysis of envisaged OAE mineral deployment in low-energy environments.

**Table 2 Advantages and disadvantages of coastal marine mineral addition on the continental shelf (low energy environment) compared to the surf zone (high-energy environment).**

| | Low-energy environment (continental shelves) | High-energy environment (surf zone) |
|---|---|---|
| **Mineral dissolution enhanced by** | metabolic processes<br><br>bioturbation | wave actions, tides and currents |
| **Mineral size** | medium (µm-range) | large (mm-range) |
| **Grinding** | mechanically, on land | in-situ by wave actions, water movement |
| **Risk of secondary mineral formation** | high, due to saturation levels in pore fluids | low, due to dilution by ambient seawater |
| **Background rain rate of** | detrital minerals; high risk of fast alkaline mineral burial<br><br>POC; high rates may enhance mineral dissolution | low impact |
| **Risk of mineral relocation** | medium, strong bottom currents may transport minerals to offshore depocenters | high, transport by wave action and currents out of turbulent zone |
| **TA detection, challenged by** | water depths - logistical difficulties | high dilution factors with seawater |





| | large seawater alkalinity background | turbulent waters, impacting stationary sensor systems |
|---|---|---|
| **Air-sea exchange** | enhanced by benthic-pelagic coupling | instantaneous |
| | enhanced by currents, upwelling | |

### 3.7.2 Considerations for best research practices

*Quantity of deployed mineral:* As the effects on the ecosystem by local alkalinity enhancement are still subject of current investigation, care should be taken when adding minerals to the seafloor. The risk of smothering flora, sessile organisms and clogging of burrows with mineral particles can be minimized by avoiding large deposits in the target area. It is currently

unknown whether locally enhanced TA increases in sediment porewaters driven by oversupply of minerals might be detrimental to certain organisms or produce a shift in the microbial community, potentially affecting mineral dissolution rates. Therefore, dispersed mineral distribution is desirable by, for example, sprinkler systems. Post-deployment channelling of minerals to depocenters by bottom currents might be unavoidable for fine-grained particles. To avoid an accumulation of undissolved minerals on the seafloor, potentially negatively affecting marine ecosystems, care must be taken in assessing the

quantity added. An upper limit of long-term mineral addition may be scaled to the local annual POC rain rate depending on the stoichiometry of $CO_2$ sequestration by the relevant mineral, if the $CO_2$ released from carbon respiration is quantitatively consumed by benthic weathering. This also assumes that mineral dissolution is tightly coupled to the remineralization of organic matter, which is unlikely to be universally the case. This question is currently being addressed in benthic mesocosm experiments (see chapter 4.4) and still requires verification in field trials. For mafic minerals such as olivine, further unwanted

side effects may arise due to release of heavy metals that may be toxic to marine organisms at higher concentrations (e.g., Ni). Testing the potential accumulation of metals in locally sourced sediment cores amended with minerals under laboratory conditions is recommended.

*Secondary precipitates:* The precipitation of secondary minerals either as discrete grains or on the surface of the added alkaline

mineral, decreasing its effective surface area, will hamper the efficiency of $CO_2$ removal. The conditions under which secondary mineral formation on the seafloor are favourable are mostly unknown. Identification of reaction pathways and rates will require highly precise monitoring analytics due to the large background seawater concentrations of chemical tracers. State-of-the art approaches currently include the measurement of stable Si isotopes in sediment pore fluids as a tracer for secondary phyllosilicate precipitation, and Ca isotopes for carbonate precipitation. These analyses are non-routine in most laboratories,

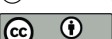

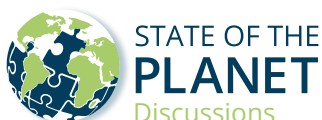

costly and labour intensive, yet must be repeated at regular intervals to correctly attribute observations to secondary mineral precipitation.

*Methods of detection:* Arguably the biggest challenge is to quantify and monitor $CO_2$ sequestration related to mineral addition on the seafloor. In low-energy deeper waters, the use of benthic chambers is an option to monitor the carbonate system over

discrete time intervals. However, artifacts such as changing redox conditions within the chamber due to oxygen depletion need to be considered and if possible, compensated for in situ. Regular sampling is also needed. In the shallow and easily accessible high-energy environment, regular water sampling is unlikely to detect alkalinity increase or $CO_2$ drawdown in this highly diluted and well-mixed environment. Long-term monitoring using autonomous chemical sensor platforms for detection of changes in pH, p$CO_2$ and TA (Sonnichsen et al., 2023) is an option, but highly challenging due to large fluctuations over

timescales ranging from hours to years. Keeping track of the mass of mineral gains over time may yield more robust results.

**Conclusions**

OAE as a potential solution to combat ocean acidification and remove $CO_2$ from the air continues to show great promise. This chapter has delved into technical aspects of the various technologies, and considerations for best practices in their research and development. Although challenges remain, such as cost-effectiveness and minimizing environmental impacts, pilot projects

have begun to demonstrate the feasibility of deploying various OAE techniques in relevant operational environments. Continued innovation and collaboration among scientists, engineers, and policymakers will be crucial in refining these technologies further. While initial experiments have been successful on a smaller scale, significant challenges lie in deploying these techniques on a global level. Addressing logistical complexities, ensuring proper monitoring and regulation, and securing necessary funding are imperative for successful scaling. Additionally, considering regional variations and selecting appropriate

sites for implementation will be vital for maximizing the efficiency and effectiveness of OAE projects. The identification and implementation of best practices are essential for the success of OAE initiatives. Conducting comprehensive environmental impact assessments, employing adaptive management strategies, and promoting transparency and public engagement are crucial steps in ensuring responsible deployment. Learning from past experiences, both positive and negative, will help refine the methodologies and minimize any unintended consequences. It is crucial to approach this strategy with careful

consideration, ensuring technology readiness, addressing scaling challenges, and implementing best practices.

**Appendices**

Definitions and common abbreviations
AWL – accelerated weathering of limestone



CDR – carbon dioxide removal

CCS – carbon capture and storage, specifically where $CO_2$ is concentrated from waste streams

DIC – total dissolved inorganic carbon

OAE – ocean alkalinity enhancement

$\Omega_{cal}$ – calcium carbonate (calcite) saturation state

$pCO_2$ – partial pressure of $CO_{2g}$

pH – negative logarithm of the hydrogen ion activity, a measure of acidity

TA – total alkalinity, the capacity of a solution to neutralize acid

**Author Contributions**

MDE, SG, and PR scoped and edited the contents of the chapter. MDE contributed to section 3.2, SG contributed to section 3.7, PR contributed to sections 3.4 and 3.5, LB contributed to section 3.5, JC contributed to section 3.1 and the conclusions, AD contributed to section 3.7, SF contributed to sections 3.1 and 3.4, PG contributed to section 3.7, OH contributed to section 3.4, CRL contributed to section 3.6, GHR contributed to section 3.2 and 3.3, JR contributed to section 3.6.

**Competing Interests**

MDE is Co-Founder and Chief Scientific Advisor to Ebb Carbon, Inc. GHR is Co-founder and Chief Technology Officer at Planetary Technologies, Inc. OH is a Research and Development Scientist for Origen Carbon Solutions.

**Acknowledgements**

PR acknowledges funding from the European Union's Horizon 2020 Research and Innovation Program under grant 869357 (project OceanNETs: Ocean-based Negative Emission Technologies analyzing the feasibility, risks, and co-benefits of ocean-
based negative emission technologies for stabilizing the climate), and philanthropic funding from the ClimateWorks Foundation and Climate Pathfinders Foundation. MDE acknowledges funding from the Grantham Foundation for the Protection of the Environment. SG acknowledges funding from the German Federal Ministry of Education and Research (Grant No 03F0895) Project RETAKE, DAM Mission "Marine carbon sinks in decarbonization pathways" (CDRmare).





This is a contribution to the "Guide for Best Practices on Ocean Alkalinity Enhancement Research". We thank our funders the ClimateWorks Foundation and the Prince Albert II of Monaco Foundation. Thanks are also due to the Villefranche Oceanographic Laboratory for supporting the lead authors' meeting in January 2023.

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
