# Peer review of "Assessing the technical aspects of ocean alkalinity enhancement approaches"

_State of the Planet, 2023_

## Referee Comment (RC1)

**Reviewer Comments:**

- The authors should indicate that ocean-CDR/OAE has the potential to compensate for CO2 degassing associated with large-scale removal of CO2 from the atmosphere. This makes ocean-CDR one of the only pathways that delivers CO2 removal with no CO2 release consequences ensuring that it is one of the most (the most?) resource efficient means for large scale CDR.

- The figure number needs to be corrected: *"The two primary electrochemical processes used to generate alkalinity from brine are electrolysis (O'Brien, Bommaraju, and Hine, 2005, pp.31-34) and electrodialysis (Strathmann, 2011, pp.163-167), as shown in Fig. 3.3."* ***Figure 3***

- Regarding Figure 3: It is not always necessary to have pretreatment, and to segregate the brine into divalent-rich, and -poor streams. The process flow could be shown more generally, and it could be indicated that many possible configurations exist.

- The authors may wish to consider the following two publications which are relevant to one particular approach for ocean-based CDR, but which contain commentary of relevance to many ocean-based CDR approaches:
    a. https://pubs.acs.org/doi/full/10.1021/acssuschemeng.0c08561
    b. https://pubs.acs.org/doi/full/10.1021/acsestengg.3c00004

- The authors are correct in that CO2 is released in a static seawater system wherein CaCO3's precipitation is induced. However, when CaCO3 precipitates under conditions of constant alkalinity generation (i.e., in the presence of a constant pH pump as proposed via the La Plante et al. process), as noted in the second reference above – no CO2 is released. Furthermore, the process of La Plante et al. is not based only on CaCO3 precipitation. Rather, CaCO3 precipitation is the secondary contributor to CDR, the primary contributor being Mg(OH)2 derived alkalinity. As such, the second sentence is inaccurate and should be corrected for content, and for its indication regarding efficiency which is not assessed herein. "In a second version of this approach, the NaOH(aq) is added to seawater to remove CO2 as CaCO3(s), with additional NaOH(aq) then added to restore this lost alkalinity and draw CO2 from the air to replace the removed CO2 (de Lannoy et al. 2018; Eisaman et al. 2018; La Plante et al. 2021). *The precipitation of CaCO3(s) releases CO2, making this second version relatively inefficient from a CO2-removal perspective, but may be pursued if other considerations such as ease of verification outweigh this inefficiency."*

- Line 259: reactions at anode and cathode are interchanged. H2 is produced at the cathode and Cl2/O2 at the anode

- Line 263: LaPlante (2023) discusses the approach described here to neutralize the acidic efflux of an electrolyzer using the acid-neutralization capacity of different solutes. It would be appropriate to cite this work.

- It is worth mentioning that the development of efficient and durable oxygen selective electrodes would be an important direction to make seawater electrolysis more feasible –

because otherwise the large-scale production and handling of gaseous chlorine (in addition to acid) could make standard electrolysis processes difficult to scale.

- Many figure numbers are written as 3.x. These should be updated/corrected.

---

## Author Response (AR2)

**Collected Responses to the Editor and Reviewers**
**Assessing the technical aspects of ocean alkalinity enhancement approaches**

We thank the Editor for his helpful comments, and are happy to include the following response.

**Editor's Comment:** Title: OAE or Ocean Alkalinity Enhancement? I suggest the latter as the abbreviation has not been introduced.
**Response**: Good point, this was a hang up from when we thought it might be presented as a chapter in a book. We've amended.

**Editor's Comment**: Fig. 1: <100 m depth or better "above mixed layer depth" ? 100 m seems an arbitrary number.
**Response**: Yes, but its illustrative. We've added 'e.g.,'

**Editor's Comment**: Line 50: I suggest adding the Fakharee reference here.
**Response**: Now added thanks.

**Editor's Comment**: Line 126 and 132: Add dates of these field trials in brackets as "currently" will be unclear for readers in the future.
**Response**: We've amended with temporal information

**Editor's Comment**: Line 193: Air-sea eq. timescales are weeks to centuries (He and Tyka, 2023 could be cited here)
**Response**: Good point, amended.

**Editor's Comment**: Line 201: (citation)?
**Response**: Good spot, Reference now added

**Editor's Comment**: Line 205: I suggest referring to Schulz et al (this guide) here as this chapter contains a thorough discussion on eta.
**Response**: Reference now added

**Editor's Comment**: Line 379: Albright et al did not necessarily show increased calcification but their results could equally well come from reduced dissolution. I suggest referring to a more physiological study here, if you want to make this point on increased biotic calcification.
**Response**: Good point, Gore et al., showed increased biotic calcification, which is already referenced.

**Editor's Comment**: Line 426: alkalinization or alkalinity enhancement for consistency?
**Response**: The point specifically is referring to ocean pH adjustment, which is not the same as OAE, so modified slightly to suit but removing alkalinization to avoid confusion.

**Editor's Comment**: Line 711: You use Orr et al., 2005 for the same statement above, maybe

be consistent.

**Response**: this occurs from two different authors, I believe the Sarmiento and Gruber reference is as good as Orr et al, we might want to cite both in parallel for places in the manuscript.

**Editor's Comment**: Line 723: The Glass et al reference is incorrect here as they have not tested mineral additions, but looked exclusively on Ni, mostly from a paleo-perspective. I am unaware of any published studies that have tested mineral additions, but here is a published pre-print. https://egusphere.copernicus.org/preprints/2023/egusphere-2023-2120/

**Response**: Thank you for the reference. We removed Glass et al. and added the Guo et al. reference.

**Editor's Comment**: Line 729: "Less clearly" suggests that we know much more about Ni than about the other TMs, but I have strong doubts. Suggest rewording so that it does not appear we know already much more about Ni impacts.

**Response**: Agreed, we changed the sentence to 'Other trace metals present in alkaline minerals, including Cu, Cd, Cr, or other heavy metals, might impose further ecotoxicological effects, depending on their concentration in the minerals transferred to the water column or sediment (Simkin & Smith, 1970, Beerling,2017).

**Editor's Comment**: Line 743: I disagree with this statement on dilution as the open ocean has lower TM concentrations and may in fact be more affected. Your choice if you want to leave this statement as is but most studies I am aware of suggest that TM effects are strong offshore where TMs are usually very low. (consider for example all the iron fertilization work).

**Response**: This is a good point, in a way, iron can't be compared with the rather toxic TMs as it's a nutrient and organisms take it up actively and have mechanisms to incorporate it. The whole question would however be about tolerance or organisms, this might certainly vary with the diversity present, the ecotypes we find etc. Therefore, the original statement is in a way meaningless as it would assume a steady community. This is of course never right, therefore, I changed the statement to 'The potential difficulty arising from introducing toxic compounds would vary strongly depending on the respective ecosystem's tolerance to those compounds.'

**Editor's Comment**: Line 747: pH 9 is not really within natural bounds but a rarity to ultra-eutrophic systems (e.g. heavily polluted shallow bays). I suggest re-considering this number.

**Response**: We changed the statement to the upper limit found in ocean surface waters; it now reads:

'Furthermore, seawater pH should be kept within a natural seawater range (pH $\leqslant$ 8.2; Pedersen and Hansen,2003).'

**Editor's Comment**: Line 737-759: I found these two paragraphs comparatively weak and suggest considering deleting them. They appear off-topic to the chapter and contain rather out-dated information. But this is just a suggestion.

**Response**: We respectfully disagree and would need to receive more specific details to be able to follow.

**Editor's Comment**: Line 817: Montserrat and Meysman did not show this (they may have speculated) but Flipkens et al., 2023 recently did.
https://www.sciencedirect.com/science/article/pii/S0016703723004349.
**Response**: Thank you for pointing this out, indeed Montserrat & Meysman never proved the hypothesis to be true. We changed the citation to Flipkens et al. (2023).
Response to Anonymous Reviewer #1 Comments on "Assessing technical aspects of ocean alkalinity enhancement approaches"

We thank the referee for their careful review of the manuscript and for their constructive comments. All comments are addressed point by point below.

1. **Reviewer Comment:** "The authors should indicate that ocean-CDR/OAE has the potential to compensate for CO2 degassing associated with large-scale removal of CO2 from the atmosphere. This makes oceanCDR one of the only pathways that delivers CO2 removal with no CO2 release consequences ensuring that it is one of the most (the most?) resource efficient means for large scale CDR."

   **Response:** Excellent point. The following text has now been added at lines 967-968 in the Conclusions section: "OAE's unique potential among CDR approaches to compensate for $CO_2$ degassing from the ocean resulting from large-scale atmospheric $CO_2$ removal makes it an especially valuable approach worthy of continued pursuit."

2. **Reviewer Comment:** "The figure number needs to be corrected: "The two primary electrochemical processes used to generate alkalinity from brine are electrolysis (O'Brien, Bommaraju, and Hine, 2005, pp.31-34) and electrodialysis (Strathmann, 2011, pp.163-167), as shown in Fig. 3.3." ***Figure 3"

   **Response:** Thank you, this has been corrected to read: "The two primary electrochemical processes used to generate alkalinity from brine are electrolysis (O'Brien, Bommaraju, and Hine, 2005, pp.31-34) and electrodialysis (Strathmann, 2011, pp.163-167), as shown in Fig. 3."

3. **Reviewer Comment:** "Regarding Figure 3: It is not always necessary to have pretreatment, and to segregate the brine into divalent-rich, and -poor streams. The process flow could be shown more generally, and it could be indicated that many possible configurations exist."

   **Response:** Good point, thank you for catching this. The figure has been revised to show pretreatment and divalent separation as optional and the following text has been added to the caption of Fig. 3: "Both pretreatment and the separation of brine into divalent rich and divalent lean streams is optional and is not performed in all processes."

4. **Reviewer Comment:** "The authors may wish to consider the following two publications which are relevant to one particular approach for ocean-based CDR, but which contain commentary of relevance to many ocean-based CDR approaches:
   a. https://pubs.acs.org/doi/full/10.1021/acssuschemeng.0c08561
   b. https://pubs.acs.org/doi/full/10.1021/acsestengg.3c00004

   **Response:** La Plante et al, 2021 was already cited in the original manuscript. Thank you for pointing us to the recent La Plante et al., 2023 paper. Both are now cited. An additional sentence was added (lines 253 – 255) that reads: "A third version of this

approach relies primarily on the precipitation of $Mg(OH)_2$, in addition to the precipitation of some $CaCO_3$, and prevents release of $CO_2$ in the process of $CaCO_3$ precipitation by generating alkalinity at a sufficiently high rate to keep the pH at a constant target value (La Plante et al. 2021; La Plante et al., 2023)."

5. **Reviewer Comment:** "The authors are correct in that CO2 is released in a static seawater system wherein CaCO3's precipitation is induced. However, when CaCO3 precipitates under conditions of constant alkalinity generation (i.e., in the presence of a constant pH pump as proposed via the La Plante et al. process), as noted in the second reference above – no CO2 is released. Furthermore, the process of La Plante et al. is not based only on CaCO3 precipitation. Rather, CaCO3 precipitation is the secondary contributor to CDR, the primary contributor being Mg(OH)2 derived alkalinity. As such, the second sentence is inaccurate and should be corrected for content, and for its indication regarding efficiency which is not assessed herein. "In a second version of this approach, the NaOH(aq) is added to seawater to remove CO2 as CaCO3(s), with additional NaOH(aq) then added to restore this lost alkalinity and draw CO2 from the air to replace the removed CO2 (de Lannoy et al. 2018; Eisaman et al. 2018; La Plante et al. 2021). The precipitation of CaCO3(s) releases CO2, making this second version relatively inefficient from a CO2-removal perspective, but may be pursued if other considerations such as ease of verification outweigh this inefficiency.""

    **Response:** Thanks for this clarification. The existing sentence was kept as is, but the La Plante et al, 2021 reference was removed from that sentence. An additional sentence was added (lines 253 – 255) after the sentence in question that reads: "A third version of this approach relies primarily on the precipitation of $Mg(OH)_2$, in addition to the precipitation of some $CaCO_3$, and prevents release of $CO_2$ in the process of $CaCO_3$ precipitation by generating alkalinity at a sufficiently high rate to keep the pH at a constant target value (La Plante et al. 2021; La Plante et al., 2023)."

6. **Reviewer Comment:** "Line 259: reactions at anode and cathode are interchanged. H2 is produced at the cathode and Cl2/O2 at the anode"

    **Response:** Thanks for catching this error. It has been corrected.

7. **Reviewer Comment:** "Line 263: LaPlante (2023) discusses the approach described here to neutralize the acidic efflux of an electrolyzer using the acid-neutralization capacity of different solutes. It would be appropriate to cite this work."

    **Response:** Agreed, thanks. La Plante et al., 2023 is now cited at line 277 as: "Such acids (including the hydrochloric acid described in the previous section) can be reacted with alkaline minerals to produce more neutral metal salts and water (La Plante et al., 2023)."

8. **Reviewer Comment:** "It is worth mentioning that the development of efficient and durable oxygen selective electrodes would be an important direction to make seawater

electrolysis more feasible – Page 2 because otherwise the large-scale production and handling of gaseous chlorine (in addition to acid) could make standard electrolysis processes difficult to scale."

**Response:** Agreed. A sentence was added at line 165 that reads: "The development of efficient and durable oxygen selective electrodes is critical to making seawater electrolysis more feasible (La Plante et al., 2023)."

9. **Reviewer Comment:** "Many figure numbers are written as 3.x. These should be updated/corrected."

   **Response:** Thank you. All figure references have been corrected.

Response to Prof. Justin Ries Comments on "Assessing technical aspects of ocean alkalinity enhancement approaches"

We thank the referee for their careful review of the manuscript and for their constructive comments. All comments are addressed point by point below.

1. **Reviewer Comment:** "Although highly informative, the manuscript is quite long and, at times, somewhat redundant. For example, the issue of trace element release is brought up in numerous sections, sometimes providing little additional information relative to the prior section. One way to shorten the length of the manuscript and increase its concision would be to combine all the sections on trace element release into a single section at the end (or the beginning), and then reference that section for the various sections addressing different methods of OAE. The same could be done with ecological impacts."

   **Response:** Thank you for the comment. Trace metals are considered in four sections electrochemical approaches, AWL, pelagic and benthic addition. While there is some similarity in these various descriptions, the causes and implications of metal release from these approaches varies and it would not be appropriate to combine them into a generic section, and may not shorten the text significantly. Ecological impacts also vary for each of these approaches.

2. **Reviewer Comment:** "would also encourage the authors to take this opportunity to clarify the matter around reprecipitation of carbonates following net alkalinity addition reducing the efficiency of OAE..."

   **Response:** Thank you, this is addressed in comments 14 and 15 below.

3. **Reviewer Comment:** Line 22: 'and' instead of 'or'

   **Response:** Changed as suggested.

4. **Reviewer Comment:** "63: 'pCO2'

   **Response:** Added partial pressure of CO2

5. **Reviewer Comment:** 65: this would work with oxides (e.g., CaO, MgO), as well, no?

   **Response:** Notionally, alkaline materials added and example hydrated lime now included.

6. **Reviewer Comment:** "Line 259: reactions at anode and cathode are interchanged. H2 is produced at the cathode and Cl2/O2 at the anode"

   **Response:** Thanks for catching this error. It has been corrected.

7. **Reviewer Comment:** "The statement that 'For these approaches to be meaningful for CDR, the concentrated CO2 used in the process must come from the atmosphere' is too stringent… The concept of marine CDR should be expanded from 'direct removal of CO2 from the atmosphere' to include the 'transfer of CO2 in the atmosphere and/or seawater into stable carbonate or bicarbonate ions in seawater', as both processes (i.e., removal of CO2 from atmosphere and/or seawater) will result in the eventual drawdown of CO2 from the atmosphere once equilibrium wrt pCO2 is re-established between these coupled systems.."

   **Response:** Good point, 'or the surface ocean' added

8. **Reviewer Comment:** "Likewise, CO2 removal from the atmosphere alone is not sufficient for CDR, as increasing the pCO2 of the atmosphere through increased CO2 emissions also drives the flux of CO2 from the atmosphere to the ocean, but surely this should not constitute marine CDR. Perhaps a more useful framing for CDR is the transfer of C from shorter residence time reservoirs (atmospheric CO2, seawater CO2, terrestrial biomass, marine biomass in mixed layer etc.) to longer residence time reservoirs (bicarbonate ion reservoir, carbonate ion reservoir, terrestrial and marine biomass transported to deep ocean and/or into marine sediments below mixed layer, etc.) (c.f., Prentice, I. C., 2001,

   **Response:** There is a range of framing to consider CDR, each with its own limitations. The removal of CO2 from the quick bio cycle to the geo cycle, while reasonable for what we are suggesting here, would exclude approaches that increase stocks of biomass.

9. **Reviewer Comment:** 70: It seems that the authors should more explicitly address the fact that calcination of CaCO3 to produce Ca(OH)2 releases CO2, which offsets more than half of the CO2 sequestration potential associated with using the produced Ca(OH)2 for OAE. They mention storing or reusing the CO2, but at Gt scale this would probably not be practical. It should also be highlighted that Mg(OH)2 is unique in this

regard bc, unlike Ca(OH)2, it is naturally occurring and can be mined, and thus does not require calcination in its production.

**Response:** We address this later in the chapter when we consider ocean liming, and it would be better to explain in detail later than include too many clauses in the summary statements here. Mg-hydroxide, while naturally occurring, does not exist abundantly in pure deposits and would need to be 'extracted/purified' from host rock.

10. **Reviewer Comment:** 101: delete 'which'

    **Response:** replaced "which" with "that"

11. **Reviewer Comment:** 115: But the technologies shown between TRL 4 and 7 in the accompanying figure are being explored by both researchers and companies, so, although the general TRL discussion is informative, not sure how relevant the 'valley of death' discussion is here.

    **Response:** Indeed, this is being explored by researchers and companies, yet it remains a risk for developing OAE approaches. The valley of death concept refers to the gap between research and translation and applies to TRL level, whether that research is being pursued in academia or industry.

12. **Reviewer Comment:** 131: Consider removing the informal reference to research being conducted at H-W University: Finally, the production and application of hydrated carbonate minerals such as ikaite has a TRL of 1, currently under investigation at the bench-scale at Heriot-Watt university examining aspects such as air stability and seawater dissolution kinetics.' If a citable reference exists for this research, then the reference can be included and discussed. Otherwise, its more of an informal communication that is probably not suited for the present contribution.

    **Response:** Agreed. Text removed, and reference is now included.

13. **Reviewer Comment:** 236: Since alkalinity is defined as the sum of all proton-neutralizing ions minus the sum of protons in a solution, it could also be argued that removing HCl (and thus protons) from seawater via 'direct ocean capture' is indeed a form of ocean alkalinity enhancement because it is removing CO2 from the atmosphere by enhancing (increasing) the alkalinity of seawater.

    **Response:** Direct ocean capture, while it might use acids and bases to facilitate the removal of CO2 gas, does not result in a net change of alkalinity. For example, direct ocean capture that strips CO2 as a gas from seawater first generates equal parts H+ and OH- from the water in seawater, resulting in no net alkalinity change. The acid is then

added to seawater to shift all DIC to the form of dissolved $CO_2$ gas, reducing alkalinity. Because the equilibrium partial pressure of 2.4 mmol DIC converted to $CO_2$ gas is approximately 0.08 atm (i.e., less than 1 atm), this $CO_2$ must be vacuum stripped from seawater. The alkalinity is now restored to the acidified and decarbonized seawater by adding the $OH^-$ generated in the first step. This simply return the alkalinity to the starting point.  The restored alkalinity results in the absorption an equal amount of $CO_2$ from the air into the seawater, restoring the $CO_2$ that was stripped from solution. In this way, direct ocean capture results in no net increase in alkalinity, but uses acid and base to cycle alkalinity in a way that uses the ocean to "pump" $CO_2$ from the air.

14. **Reviewer Comment:** 244: 'The precipitation of CaCO3(s) releases CO2'; this is a bit misleading as presently written. If the precipitation of CaCO3 is caused by the net addition of alkalinity to a solution (such as in applied OAE using metal hydroxides, or the like), it will not result in the net release of CO2. This is because the H+ generated from the calcification process are effectively neutralized by the added alkalinity. CO2 is only released on a net basis by CaCO3 precipitation when the CaCO3 precipitation is driven by an elevation in the pH and saturation state in the absence of a net increase in alkalinity, such as that driven by CO2 drawdown via photosynthesis. The idea that CaCO3 precipitation following net alkalinity addition is a net emitter of CO2 is a widely misunderstood aspect of the seawater carbonate chemistry system that I would suggest clarifying here because it is is creating confusion in the field of OAE and applied CDR

CaCO3 precipitation without net alkalinity addition: Ca2+ + HCO3 -> CaCO3 + H+ (net acid and thus net CO2 release)

[and]

15. **Reviewer Comment:** That said, precipitating CaCO3 after OAE does reduce the efficiency of CDR via OAE because it only removes 1 mole of CO2 for every 2 moles of alkalinity (CaCO3 contains 1 mole DIC per 2 moles alkalinity), while conventional OAE is believed to remove approximately 1.4 to 1.6 moles CO2 for every 2 moles of alkalinity. So the efficiency of OAE decreases by about 1/3 if the CO2 and alkalinity reprecipitate out as CaCO3, rather than staying balanced in a dissolved state.

**Response:** Agreed, we have tightened the language to state that carbonate precipitation reduces alkalinity resulting in a lower efficiency of removal or CO2 release in the case of OAE. The new text reads: "The precipitation of $CaCO_{3(s)}$ reduces alkalinity (resulting in either in lower efficiency of $CO_2$ removal per unit of added alkalinity in the case of OAE, or a release of $CO_2$ in cases where the precipitation occurs in the absence of an alkalinity addition)..." We appreciate the wider comment regarding the precipitation of CaCO3 and the impact on the carbonate system in seawater. However, this chapter considers the technical aspects of OAE and there are chapters dedicated to describing the implications of OAE on carbonate geochemistry (see Schulz et al.,).

16. **Reviewer Comment:** 338: should there be an 'and' between 'calcium' and 'bicarbonate'?

**Response:** Agreed, we have changed this as suggested.

17. **Reviewer Comment:** 438: consider elaborating on the 'flue gas' that is mentioned—is this the source of the CO2 that is being sequestered and used to drive the dissolution of the limestone?
    **Response:** Added 'This study explored the dissolution of limestone in seawater driven by high CO2 from a point source of emissions rather than concentrated from the atmosphere, which may not have the same impurities.'

18. **Reviewer Comment:** 560-566: this is very useful information wrt liming via Ca(OH)2 addition. Since several workers are also investigating OAE via Mg(OH)2 addition, is it possible to report similar recommendations for Mg(OH)2, which has the benefit of being a naturally occurring metal hydroxide that therefore, unlike Ca(OH)2, does not require calcination (assuming the Mg(OH)2 is mined, not calcined from MgCO3)?
    **Response:** Pure Mg(OH)2 deposits are not sufficiently abundant that they can be mined and used for OL. It is found in reasonable concentrations (5-10%) in ultrabasic rock, but its extraction and purification from this for OAE has not been explored.

19. **Reviewer Comment:** 625: I would recommend including aragonite, high-Mg calcite, and even dolomite in Table 1 (only pure calcite is presently included) because these forms of CaCO3 are globally abundant, some have existing mining industries built around them (dolomite), and are generally more soluble than pure calcite and, thus, potentially advantageous for OAE, ALW, etc
    **Response:** The intention of including calcite in table 1 is for comparison of Gibbs free energies to hydrated phases, not for its specific use for OAE. There would be no additional benefit in including additional anhydrous phases.

20. **Reviewer Comment:** 685: consider clarifying 'upper water column' (thus in contact with the atmosphere to allow for equilibration wrt pCO2)
    **Response:** The "upper water column" refers to the mixed layer depth, which is influenced by atmospheric processes and therefore allows for equilibration with $CO_2$. We clarified this in the text.

21. **Reviewer Comment:** 692: and dolostone (or dolomite)
    **Response:** We added this to the sentence.

22. **Reviewer Comment:** 702: again, reprecipitation of CaCO3 after non-carbonate alkalinity addition (e.g., Mg(OH)2) does not completely negate the CO2 sequestration associated with the net alkalinity addition, it only reduces its efficiency by about 1/3 (see line 244 above).
    **Response:** We agree and have added a clarification to the text.

23. **Reviewer Comment:** 858: the risk of secondary precipitation of carbonates in benthic environments outside of the tropics is relatively low; most deep-sea benthic

environments are of course relatively undersaturated wrt CaCO3, and even shallow shelf sediments experience net CaCO3 dissolution in temperate latitudes and higher due to undersaturation at the sediment-water interface, in part owing to respiration of CO2 from organic matter.

**Response:** We agree that organic matter degradation and the related drop in pH is a process causing carbonate dissolution and is contrary to the carbonate precipitation risk, discussed in this section. However, in coastal benthic environments microbial reactions can also induce carbonate precipitation by e.g. alkalinity increase during AOM or in conjunction with silicate dissolution, buffering the pH drop by OM degradation and by this, inhibiting carbonate dissolution. We clarified this part in lines 874-875.

24. **Reviewer Comment:** 892: May want to clarify that high rates of siliciclastic sedimentation should result in lower rates of organic matter degradation (ie., higher rates of organic matter preservation). The text here seems to suggest the opposite. It is a bit unclear whether the author is referring to rates of deposition of organic matter ('POC sedimentation'), or to rates of siliciclastic sedimentation ('detrital minerals', 'OAE minerals'). It may be worth parsing out this discussion to address different types of sedimentation separately in order to improve clarity of this section.

    **Response:** Thank you for pointing this out. This section and the distinction between POC and detrital mineral precipitation rates was revised in lines 902-907.

25. **Reviewer Comment:** 920: should 'favourable' be 'favoured'; this sentence is a bit unclear as written.

    **Response:** The sentence was re-written to increase clarity.

26. **Reviewer Comment:** 921: isotope tracers are not the only way to quantify secondary precipitation; direct measurement of secondary precipitates in sediment samples may be a more straightforward and accessible method worth mentioning. This sections seems to be overstating the difficulty of characterizing secondary precipitation within benthic sediments.

    **Response:** Thank you for highlighting this bias towards isotope tracers. We fully agree that more methods are available to identify secondary precipitates and we revised the section accordingly.